# Fully bioresorbable hybrid opto-electronic neural implant system for simultaneous electrophysiological recording and optogenetic stimulation

Myeongki Cho[1,11], Jeong-Kyu Han[2,11], Jungmin Suh[1,11], Jeong Jin Kim [3], Jae Ryun Ryu[4], In Sik Min [1], Mingyu Sang[1], Selin Lim[1], Tae Soo Kim[1], Kyubeen Kim[1], Kyowon Kang[1], Kyuhyun Hwang[5], Kanghwan Kim[2], Eun-Bin Hong[6], Min-Ho Nam [6], Jongbaeg Kim [5], Young Min Song [7], Gil Ju Lee [3] ✉, Il-Joo Cho [8,9] ✉ & Ki Jun Yu [1,10] ✉

Bioresorbable neural implants based on emerging classes of biodegradable materials offer a promising solution to the challenges of secondary surgeries for removal of implanted devices required for existing neural implants. In this study, we introduce a fully bioresorbable flexible hybrid opto-electronic system for simultaneous electrophysiological recording and optogenetic stimulation. The flexible and soft device, composed of biodegradable materials, has a direct optical and electrical interface with the curved cerebral cortex surface while exhibiting excellent biocompatibility. Optimized to minimize light transmission losses and photoelectric artifact interference, the device was chronically implanted in the brain of transgenic mice and performed to photostimulate the somatosensory area while recording local field potentials. Thus, the presented hybrid neural implant system, comprising biodegradable materials, promises to provide monitoring and therapy modalities for versatile applications in biomedicine.

Advances in neural implant technology, which directly connects external devices with the nervous system to modulate or monitor neural activities, have provided not only a route to functional identification of brain regions, but also, a promising engineering approach to the clinical treatment of incurable neurodegenerative diseases including Parkinson's disease, Alzheimer's disease, and epilepsy[1,2]. In the early stage, neuroscientists focused on using electrode arrays to record single or multiple neural activities, paving the path for the implementation of brain mapping and brain–machine interfaces[3–5]. However, to carry out effective pathological diagnosis and treatment of the brain, it is essential to move beyond mere neural recording and integrate monitoring of various factors alongside the application of suitable stimulation[6–8]. In this regard, one of the most significant

aspects of progress is the emergence of optogenetics, a technique that enables controlling target neurons with millisecond-order temporal accuracy control through optical stimulation, by expressing photo-active microbial opsin proteins in the desired nerve cells[9,10]. Optogenetics, unlike electrical stimulation, not only enables selective expression of opsin in specific neurons or multiple opsins in a single cell for independent manipulation of excitation and inhibition but also minimizes signal interference during electrophysiological recordings[11–13]. Consequently, there have been growing efforts to address currently incurable neurological diseases based on the optogenetic approach.

Comprehensive understanding of the sophisticated and intricate interplays among distinct neural populations in the brain requires

innovative, implantable neural interfaces capable of manipulating and interrogating target neurons across multiple locations with high spatiotemporal resolutions[14]. Thus, the development of micro-neural implant electronics (μ-NIE) based on microelectromechanical systems has led to their extensive use in various applications for pathological research on neurological disorders over the past few decades by incorporating advanced optogenetic techniques[15–19]. Despite the potential of neural implants as promising tools in neurosciences, the μ-NIE are manufactured using commercially available inorganic materials, resulting in significant mechanical mismatches between the device and brain tissue. When μ-NIE based on silicon or metals with high Young's modulus over 100 GPa are implanted in the brain tissue having only 3 kPa of Young's modulus, it can cause acute tissue damage, as well as inflammatory responses, which forms glial scars and loss of neurons[20,21]. To minimize immune responses and damage to the surrounding tissues after implantation, various neural implants have attained high biocompatibility through engineering improvements to the materials and structures of the device, based on nanomaterials (e.g., nanomembranes, nanowires, and nanoparticles)[22–24], soft polymers[25,26], and flexible thin-film structures[27–29]. The flexible and soft neural implants exhibit better biocompatibility compared to traditional μ-NIE to enable long-term device implantation of the device and also increase device performance by forming conformal contacts between the device and curved surfaces of tissue[30,31]. Despite the improved biocompatibility of implanted devices, secondary surgeries causing additional damage to the regenerated tissues and increasing risk of complications for the patient were still necessary to remove implanted devices after their expected periods of operation.[32].

Emerging classes of bioresorbable neural implants that operate for specific durations while maintaining optimal performance and then decompose into untraceable small elements that are absorbed or excreted from the body provide promising solutions to the challenges associated with implanted device removal[33]. Recently, various transient neural electronics have been studied for electrophysiological recordings (e.g., brain mapping)[34], biophysical monitoring (temperature, pressure, blood flow, etc.)[35–37], and sensing chemical factors via biomarkers (e.g., dopamine)[38]. In addition, implantable devices that electrically stimulate specific nerve systems (e.g., spinal cord and peripheral nerve stimulation)[39,40] or optically stimulate genetically manipulated neurons (e.g., optogenetics)[41] have been developed. Thus, by adopting approaches incorporating fundamentally distinct biodegradable materials[42], bioresorbable implant systems could potentially overcome the limitations of current devices and offer opportunities in various biomedical fields via the fabrication of devices with biocompatible by-products and avoiding secondary device extraction surgeries. However, despite the substantial research on bioresorbable neural implants, the majority of devices focus on single functions, such as recording, monitoring, or stimulation[33,42]. The challenge in designing multifunctional bioresorbable neural implants lies in achieving the monolithic integration of various materials and functions within a single device, which causes unwanted interactions or mechanical failure of the device when combining materials with different mechanical and electrical properties[43,44]. Moreover, the incorporation of multiple functions such as monitoring and stimulation requires sophisticated design and fabrication techniques that increase the complexity of the device[45]. Therefore, to realize the potential benefits of fully bioresorbable neural implants in future biomedical applications, additional developments are needed for multifunctional integrated devices that are capable of both continuous monitoring of brain lesions for diagnosis and clinical treatments through selective and accurate stimulations simultaneously.

In this study, we introduce the materials and design of a fully bioresorbable flexible neural implantable hybrid opto-electronic system for simultaneous electrophysiological recording and optical stimulation. The flexible and soft bioresorbable hybrid device establishes direct optical and electrical interfaces with the cerebral cortex, allowing selective optical stimulation and recording of neural activities in designated areas. The electrical interface of the system is constructed using an emerging class of biodegradable materials with high electrical conductivity founded on single crystalline silicon. Additionally, the optical interface comprises a waveguide with an external light source (laser ~460 nm wavelength) based on a soft, biodegradable poly-(lactic-co-glycolic acid) (PLGA) copolymer that triggers total internal reflection (TIR) at the interface with the cerebrospinal fluid. Immersed in biofluids, the device is composed exclusively of biodegradable materials that dissolve through chemical reactions, including hydrolysis, producing noncytotoxic end products that are absorbed or excreted[46–48]. This hybrid system with well-organized, integrated electrical and optical interfaces is optimized to minimize light transmission losses on the target brain region through adjustable parameters of the waveguide tip angle and electrode grid design. In addition, to minimize interference from photoelectric artifacts in the recorded electrophysiological signals caused by photovoltaic effects of light absorption by the Si layer during optogenetic stimulation, the Mo/Si bilayer electrode structure was adopted. The Mo layer prevents light absorption on the Si surface, and the Mo/Si electrode was verified in both brain-mimicking hydrogel and the cerebral cortex of wild-type mice[49]. In the animal model evaluation, the bioresorbable hybrid system was implanted in the brain of Thy-1: Channelrhodopsin-2 (ChR2) transgenic mice for the recording of the evoked local field potentials (LFPs) including spontaneous activity and induced seizure-like spike activity, in anesthetized mice captured with a conventional flexible electrocorticogram (ECoG) electrode array as the reference. Furthermore, these chronically implanted bioresorbable electronics enabled optogenetic stimulation of the somatosensory area of the cerebral cortex using 460 nm blue light, simultaneously recording the evoked LFPs over 2 weeks, and completely biodegrading within 8 weeks. Additionally, tests conducted both in vitro and in vivo confirmed high biocompatibility for the device.

## Results
### Fully bioresorbable hybrid opto-electronic neural implant system
Figure 1a shows the schematic illustration of the overall system with an electrode array for neural recording integrated onto waveguides for optical stimulation. This fully biodegradable neural implant system includes a waveguide based on PLGA, a biodegradable polymer, and an electrode array comprising $SiO_2$, as an insulating layer, and Mo/Phosphorus (P)-doped Si (n-type Si) as a conductive layer for electrophysiology. Here, the front end of the waveguide is combined with the core of an elaborately cut step-index multi-mode optical fiber (core diameter: 105 μm) to deliver light of 460 nm generated from a laser source. The terminal pads, at the front end of the electrode array consisting of Si that is exposed to $SiO_2$, are interconnected with the anisotropic conductive film (ACF) cable for neural recording. Figure 1b shows a photograph of the completed device integrated with a six-channel electrode array and waveguide. The light from the laser source is transmitted through the waveguide to stimulate the target area on each electrode while simultaneously recording neural activity. Figure 1c displays a photograph in which the flexible device is wound on a glass rod with a bending radius of 4 mm. The device constructed with a nanomembrane electrode array transferred to a PLGA waveguide substrate is soft and flexible, minimizing the modulus mismatch between tissue and implant and allowing conformal contact with the tissue (Supplemental Fig. S1). As shown in Fig. 1d, each electrode has a grid shape (line width: 10 μm, fill factor: 42%) measuring site in area of $280 \times 280$ μm$^2$ to record neural activity in the cerebral cortex. The electrode consists of a bilayer of biodegradable conductor Mo and P-doped Si, and the tissue is in contact with the exposed P-doped Si surface. In Fig. 1e, the electrode is precisely aligned and integrated with

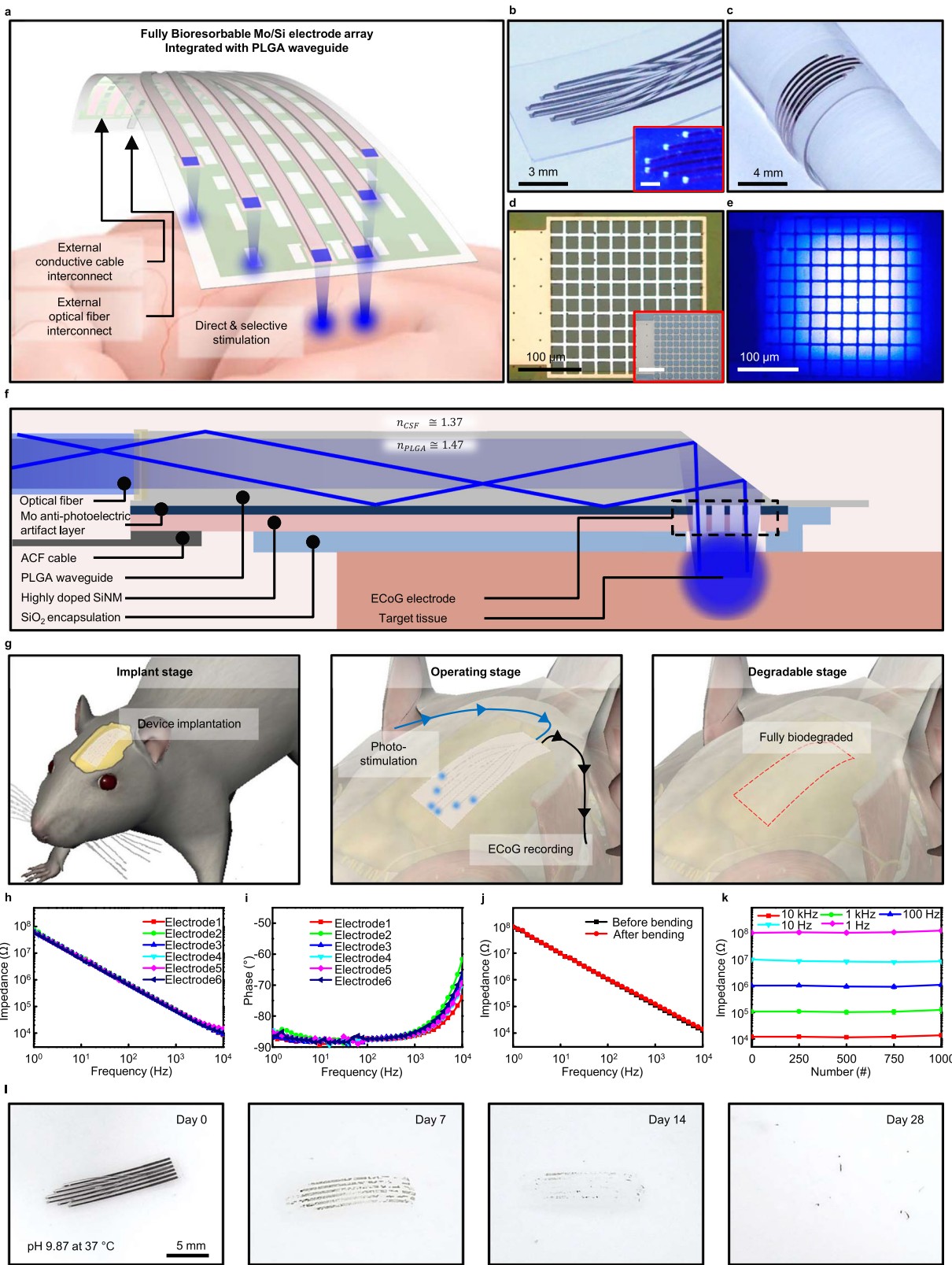

the waveguide tip to simultaneously perform photo-stimulation and electrophysiological recordings at the target site. Figure 1f presents a detailed cross-sectional view of the implanted device for optogenetic stimulation by the light-guiding to the target tissue, which is the same site as the ECoG Mo/Si electrode. The electrode array is constructed on a PLGA-based transparent substrate (20 μm) in which embossed waveguides (100 μm) are shaped through soft lithography

(Supplemental Fig. S2). The conductive electrode is composed of a Mo and highly P-doped Si nanomembrane bilayer (300/300 nm). At the same doping concentration, n-type Si comprising the electrode has a relatively lower resistivity due to its electrons, the major carriers, having higher mobility compared to the holes in p-type Si. This results in minimized low-frequency signal loss in electrodes and transmission lines. And the Mo layer is integrated with the PLGA substrate by

**Fig. 1 | Soft and flexible fully bioresorbable hybrid neural implant system constructed with a conductive Mo/Si bilayer electrocorticogram (ECoG) electrode array stacked over a biodegradable poly-(lactic-co-glycolic acid) (PLGA) waveguide. a** Schematic illustration of the fully bioresorbable neural implant system constructed with Mo/Si bilayer nanomembrane electrode array stacked on a soft and flexible PLGA waveguide for simultaneous electrophysiology and optogenetics. **b** Photograph of the biodegradable neural implant system with 6-channel ECoG electrode array and waveguide for optical stimulation. Inset is the bottom side of the device. **c** Photograph of a device wound on a glass rod of radius 4 mm. **d** Optical microscope image of the Mo/Si bilayer nanomembrane electrode for ECoG recording. Inset is the Mo layer on the backside of the electrode. **e** Optical microscope image of the 460 nm blue light emitted through the Mo/Si electrode integrated with a PLGA waveguide. **f** Schematic side-view illustration of the structure of the fully biodegradable neural implant system for simultaneous electrical recording and optical stimulation. **g** Illustrations of the main operating steps of the fully biodegradable neural implant system. The device is first implanted on the cerebral cortex (left), followed by light stimulation and recording (center), and device degradation and absorption (right). Electrochemical impedance spectra of the individual recording sites of the 6-channel neural electrode array. **h** Magnitude, **i** phase, and **j** impedance spectra of the biodegradable electrode array under flat and bent states with a 2 mm bending radius. **k** Impedance changes according to frequency during the bending endurance test with a 2 mm bending radius for 1000 cycles. **l** Photographs of the accelerated dissolution of the device by immersion in PBS with pH 9.87 at 37 °C.

transfer printing. The outermost layer of the device is made of SiO$_2$ (150 nm) which not only insulates the device from tissue but also protects it from biofluids. This inorganic thin-film encapsulation layer on the device offers higher encapsulation performance with lower water vapor transmission rate compared to organic-based encapsulation layers, ensuring stable device protection even with its thin thickness[50–53]. The detailed fabrication processes are shown in Supplemental Figs. S3–S5. The light from the laser source is transmitted through an external optical fiber coupled to the end of the waveguide of the device. The light entering the waveguide is guided through the waveguide by inducing TIRs at the interface between the cerebrospinal fluid (CSF) ($n = 1.37$) and PLGA ($n = 1.47$) (Supplemental Fig. S6). At the bottom side of the waveguide, reflection occurs at the interface between the PLGA substrate and electrode array, which minimizes an artifact of the Si electrode. Considerable absorption loss is present since reflection from metal inevitably accompanies absorption. Nevertheless, the proposed waveguide delivers a substantial power enough to stimulate neurons (Supplemental Figs. S7–S9). The guided light is transmitted to each electrode and undergoes reflections once again at the inclined waveguide tips. Photo-stimulation of the neurons is achieved by emitting light through the window of the grid-shaped electrode aligned with the waveguide. Concurrently, each electrode that is in conformal contact with the tissue records the neural activity during photo-stimulation. The electrophysiology data are recorded with a data acquisition system, which is connected to the electrode array via an ACF cable. Since all materials that constitute the device are biodegradable, as shown in Fig. 1g, the implanted device is capable of simultaneous stimulation and recording, and will eventually degrade fully in the body. The device was immersed in phosphate-buffered saline (PBS) of pH 7.4 at room temperature, and the electrical properties of the individual electrodes were characterized by electrochemical impedance spectroscopy (EIS) between 1 and 10 kHz, which is most relevant to brain activity[54]. Figure 1h and i show the impedance and phase data measurements of the individual electrodes, which were then fitted to the equivalent Randles circuit model. The value of $R_{CT}$ is 902.9 MΩ and the value of $C_{PE}$ is 1.98 μF cm$^{-2}$, and the results fit well to the equivalent Randles circuit model (Supplemental Fig. S10). Since the device needs to be bent often for implantation into the curved tissue surfaces, its mechanical stability against bending was evaluated. Figure 1j shows the EIS measurement results when the device is flat and bent over a glass rod with a radius of 4 mm, and the results show negligible differences in impedance changes. In addition, as shown in Fig. 1k, for 1000 cyclic bending tests, the impedance change of the electrode was about 9.2% at 1 Hz and 11.2% at 100 kHz, showing a change of approximately 10% over the entire frequency range. The results of an accelerated dissolution test with sequentially collected photographs of the device soaked in PBS of pH 9.87 at 37 °C are shown in Fig. 1l. Over time, Mo, Si, and SiO$_2$, which are components of the electrode, are exfoliated from the PLGA substrate into flakes and dissolved. Further, the PLGA substrate with waveguide is dissolved by hydrolysis[48]. After implantation, the overall stability of the device is ensured for a period of 3 weeks and then operation may become

restricted due to the degradation of the SiO$_2$ insulation layer (Supplemental Fig. S11). Subsequently, the device will take 2 months to completely dissolve, and the lifetime of the device can be adjusted by controlling the material parameters such as thickness, doping concentration of Si, and composition ratio of PLGA by programming manners[42].

## Device characterizations

Light transmission through the waveguide of the device is optimized to enable selective and precise light stimulation of the target area. As shown in Fig. 2a, light entering the waveguide is guided and emitted from its tip to the tissue where the electrode is located. The light transmitted from the optical fiber to the waveguide is reflected at the interface between the waveguide and the biofluid medium, and at the interface between the waveguide and the metal layer. In particular, for efficient TIR at the interface between the fiber and the biofluid interface, it is optimal to use a light emitted from the laser source with a small emission angle, as described by the following equation:

$$\theta_{in,c} = \sin^{-1}\left(\frac{n_{PLGA}}{n_{OA}}\cos(\theta_{TIR,c})\right) \text{ and } \theta_{TIR,c} = \sin^{-1}\left(\frac{n_{biofluid}}{n_{PLGA}}\right) \quad (1)$$

Based on this equation, the light emission angle of our system ($\theta_{in}$) is ~19.5° (Supplementary Note 1 and Supplemental Fig. S12). Since TIR conditions are not dependent on light polarizations (S- and P-polarization), only light sources with smaller emission angles are considered. Additionally, effective emission of the transmitted light from the waveguide tip to the external target tissue is also essential. The tilt angle of the waveguide tip and the grid pattern design of the electrodes are the dominant factors that determine the intensity of light delivered to the tissue. Figure 2b shows optical microscopic side-view images of waveguide tips tilted from 90° to 50°. Waveguides can be manufactured by freely adjusting the shapes by the soft lithography (Supplemental Fig. S13). Figure 2c and d are optical microscopic images for various line widths and fill factors of the grid electrode. The electrode is designed with a grid shape that includes optical windows to ensure efficient light transmission from the waveguide to the tissue. One group of grid electrodes has line widths of 10, 20, and 40 μm, respectively, with a fill factor of 56 %. Another group of electrodes has line widths of 20 μm each and fill factors of 63 %, 56 %, and 42 %, respectively. Figure 2e displays the optical microscopic images of waveguide tips at different angles. Each waveguide was produced with various angles of tip through a soft lithography process using a UV-curable epoxy-based master mold by UV exposure after being fixed to an inclined zig (Supplemental Figs. S14 and S15). The incident light for these images has a wavelength of 460 nm and is displayed as blue light. As the angle of the tip decreases from 90° to 50°, the intensity of the reflected light increases at the oblique surface of the tip according to Fresnel equation[55]. The inset images in Fig. 2e, which show the results of ray-tracing simulations, theoretically support the claim that the waveguide tip at a 50° angle provides the most effective delivery of

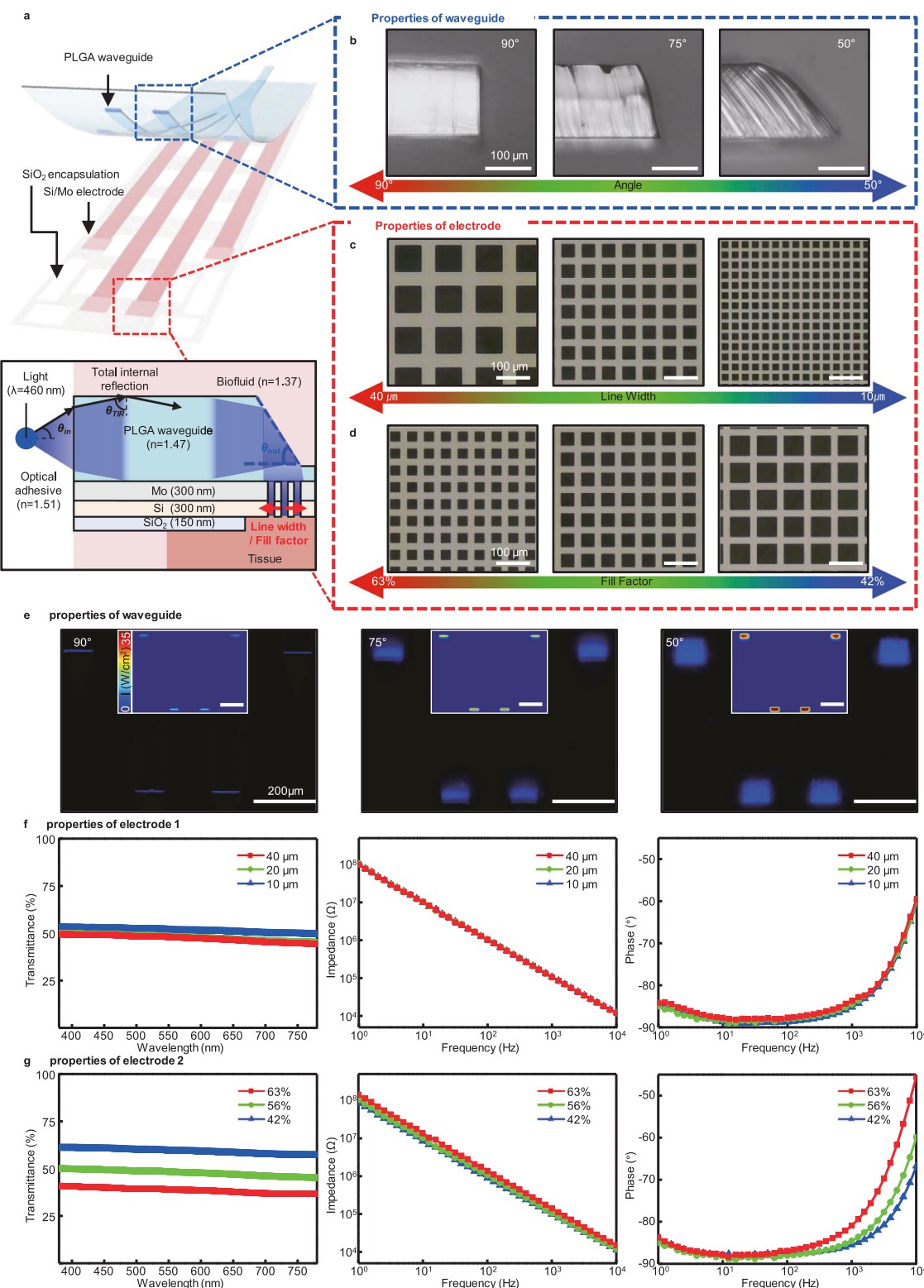

light from the source to the desired location (e.g., tissue). Quantitative comparisons were conducted between the results obtained from measurements and simulations (Supplemental Figs. S16 and S17). The "Methods" section outlines the steps taken for noise reduction and accurate comparisons of these two sets of results. Figure 2f shows the transmittance and impedance plots according to the line width of the grid electrode. Electrodes with line widths of 40, 20, and 10 μm have transmittances of 49.0 %, 49.4 %, and 52.4 %, respectively, at the wavelength of 460 nm and have transmittances of over 44.4 % in all

visible light bands. And the impedances of these electrodes are 105.2, 105.6, and 107.6 kΩ at 1 kHz, respectively, with negligible differences. Figure 2g shows the transmittance and impedance plots according to the fill factor of the grid electrode. Electrodes with 62 %, 56 %, and 42% fill factors have transmittances of 40.0 %, 49.4 %, and 60.4 %, respectively, at the wavelength of 460 nm. In addition, the electrodes show transmittances of 38 %, 44 %, and 58% in all visible light bands. Impedances of electrodes are 98.4, 105.6, and 140.7 kΩ at 1 kHz, respectively. Furthermore, the measured impedance and phase data of the

**Fig. 2 | Electrical and optical characterizations of Mo/Si electrodes and PLGA waveguide for optimizing light transmission efficiency. a** Schematic magnified illustration of the fully biodegradable neural implant system with PLGA waveguide and ECoG electrode array. The blue box indicates the tips of the waveguide, which are the light-emitting sites, and the red dotted box indicates the ECoG recording sites of the grid electrode, which is the pathway for light stimulation. Inset is a schematic diagram illustrating the pathway of light, including total internal reflection(TIR), within a waveguide. **b** Design parameters for the angle of the waveguide tip. The angles of the tips are 90° (left), 75° (center), and 50° (right). **c** Design parameters for the line width of the electrode, which are 40 μm (left), 20 μm (center), and 10 μm (right), with a fixed fill factor of 56 %. **d** Design parameters for the fill factor of the electrode, which are 63 % (left), 56 % (center), and 42 %(right), with a fixed line width of 10 μm. **e** Properties of the waveguide according to the tip angle. Optical microscope images of a waveguide emitting 460 nm light from 90° (left), 75° (center), and 50° (right) angled tips; each inset image is the ray-tracing simulation result of the biodegradable PLGA waveguide. **f** Properties of the electrode according to the line width. Visible light transmittance spectra of the electrode (left); electrochemical impedance spectra of the electrode: magnitude (center) and phase (right). **g** Properties of the electrode according to the fill factor. Visible light transmittance spectra of the electrode (left); electro-chemical impedance spectra of the electrode: magnitude (center) and phase (right).

individual electrodes were fitted to equivalent Randles circuit models (Supplemental Fig. S18). According to fill factors from 63 % to 42 %, the values of $R_{CT}$ are 568.9, 659.2, and 856.9 MΩ and the values of $C_{PE}$ are 3.14, 2.75, and 1.95 μF cm$^{-2}$, respectively. As a result, the main parameter that determines the transmittance and impedance of the electrode is the fill factor, the effective area of the electrode. The transmittance increases as the fill factor decreases, and $R_{CT}$ and $C_{PE}$ conversely depend on the effective area of the electrode because the impedance is linearly proportional to the fill factor.

## Photo-induced artifacts evaluation

Figure 3a shows the schematic illustrations of the biodegradable monolayer silicon electrode array and bilayer Mo/Si electrode array interfaced with the cerebral cortex, highlighting the photo-induced artifacts that pose major issues to the opto-electronic integration of conventional recording electrodes. A photocurrent is generated at the electrode-electrolyte interface by the photovoltaic effect (Becquerel effect)[56], which occurs when the surface of an electrode merged with an electrolyte is illuminated by light. As shown in Fig. 3b, the Fermi level of n-type Si is typically higher than the redox potential of the biofluid. When an n-type Si and biofluid form an interface, the carriers from Si are transported into the biofluid until the balance state which leads to the accumulation of surface charges and formation of a depletion region. This results in energy band bending until the Fermi level attains equilibrium[57,58]. In Fig. 3c, when light is illuminated on the surface of the electrode, photonic energy absorbed by Si causes electron–hole pairs generation that separates before recombination, leading to a photovoltage ($V_{ph}$) and a photocurrent that triggers serious contamination of the electrophysiological signals[49,59]. In the same manner, p-type Si has induced a reverse bias opposite to n-type Si (Supplemental Fig. S19). The Mo/Si bilayer electrodes have a 300 nm-thick Mo layer that blocks light from the optical interface reaching the recording site of the Si electrode, avoiding photo-induced artifacts. On the other hand, Si electrodes are directly exposed to light for neuronal stimulation, easily arising to photocurrents at the recording site. As shown in Fig. 3d, the Si nanomembrane transmits 30% of 460 nm blue light, meaning it absorbs 70 % of the light, while the Mo nanomembrane does not transmit any visible light. The electrical signals with light-induced artifacts of Si electrodes (thickness: 300 nm) and Mo/Si electrodes (thickness: 300 nm/300 nm) with 100% fill factor were measured on a substance mimicking brain tissue (0.7% agarose hydrogel). To evaluate light-induced artifacts, 460 nm laser pulses (core diameter: 105 μm, duration: 100 ms, frequency: 2 Hz) were applied to each electrode while gradually increasing the light intensity from 126.1 to 2522.5 mW mm$^{-2}$ through an optical fiber fixed perpendicular to the stereotaxis (Supplemental Fig. S20). For a more accurate experimental setup, the light intensity was measured with an optical power meter, and the electrical signals were measured with an RHD 2000 EVALUATION BOARD Version 1.0 (Supplemental Fig. S21). A detailed image of the experimental setup is shown in Supplemental Fig. S22. Figure 3e and Supplemental Fig. S23 show the electrical signals recorded from the electrode exposed to laser pulses by the intensity of light. In a lightless environment, a 50 μV level of external

noise was recorded through the electrode. However, in an illuminated environment, conspicuous artifacts were observed in the Si single-layer electrodes. The artifacts with peaks from 400 μV to 3 mV were induced as the light intensity increased from 126.1 to 2522.5 mW mm$^{-2}$ and at higher light intensities, the artifacts reached a saturation state (Supplemental Fig. S24). In contrast, when the Mo/Si bilayer electrode was illuminated at intensities below 630.6 mW mm$^{-2}$, low levels of noise without any artifacts were observed. Negligible artifacts were observed below the peak level of 200 μV with electrodes exposed to light intensities below 2522.5 mW mm$^{-2}$. Thereafter, Si and Mo/Si electrodes with 42% fill factor (line width 10 μm), allowing for the recording of ECoG while maximizing optical transmission efficiency were implanted in wild-type mice, and the photo-induced artifacts were measured using blue laser pulses (core diameter: 105 μm, duration: 100 ms, frequency: 2 Hz) (Supplemental Fig. S25). Blue light of intensity 126.12 mW mm$^{-2}$ was irradiated on the stimulation electrode, and blue light of intensity 73.15 mW mm$^{-2}$ passed through the electrode to stimulate the tissue. The control electrode was placed 1 mm away from the stimulation electrode. As shown in Fig. 3f, when the light was illuminated on the Si stimulation electrode, artifacts with a peak level of 400 μV were observed, and artifacts with a peak level of 50 μV were also observed on the control electrode owing to light scattering by the tissue. On the other hand, the Mo/Si stimulation and control electrodes did not exhibit any artifacts. When stimulating brain tissue using blue light pulses with an intensity of 2522.48 mW mm$^{-2}$, artifacts with a peak level of 200 μV were observed on the Mo/Si stimulation electrode, which was induced by the scattered and reflected light at the surface of the brain tissue, and only low-noise brain activities were recorded without artifacts at the control electrode (Supplemental Fig. S26). Furthermore, in Thy-1:ChR2 transgenic mice, optical stimulation (intensity: 63.08 mW mm$^{-2}$, duration: 30 ms, frequency: 1 Hz) was exclusively applied to a single electrode, and LFPs induced by photo-stimulation were recorded directly on the stimulated electrode (Supplemental Fig. S27). Generally, blue light intensities above 12 mW mm$^{-2}$ are needed to activate ChR2, and these results suggest that our device is capable of sufficiently stimulating and recording ChR2-expressing neurons without the interference of photo-induced artifacts[12]. The ray-tracing simulation yielded consistent results with the previous experiment (Supplemental Fig. S28). The presence of the Mo anti-artifact layer led to a remarkable reduction in photon absorption on the Si layer. Compared with the dominant absorption from the result of the Si film without the anti-artifact layer, artifacts from backscattering by the tissue are not substantial enough to have a significant impact on the recorded signal.

## Cell viability tests and immunohistochemistry analysis

To ensure the functionality and safety of the device post-implantation, a biocompatibility test was conducted on the electrode array prior to in vivo experimentation. Primary hippocampal neuronal cells were used for the test, as these are commonly used to evaluate nerve cell function in the central nervous system. As shown in Fig. 4a and b, the electrode array demonstrated high cell survival rates in both Au electrodes on polyethylene terephthalate (PET) substrate and Mo/Si

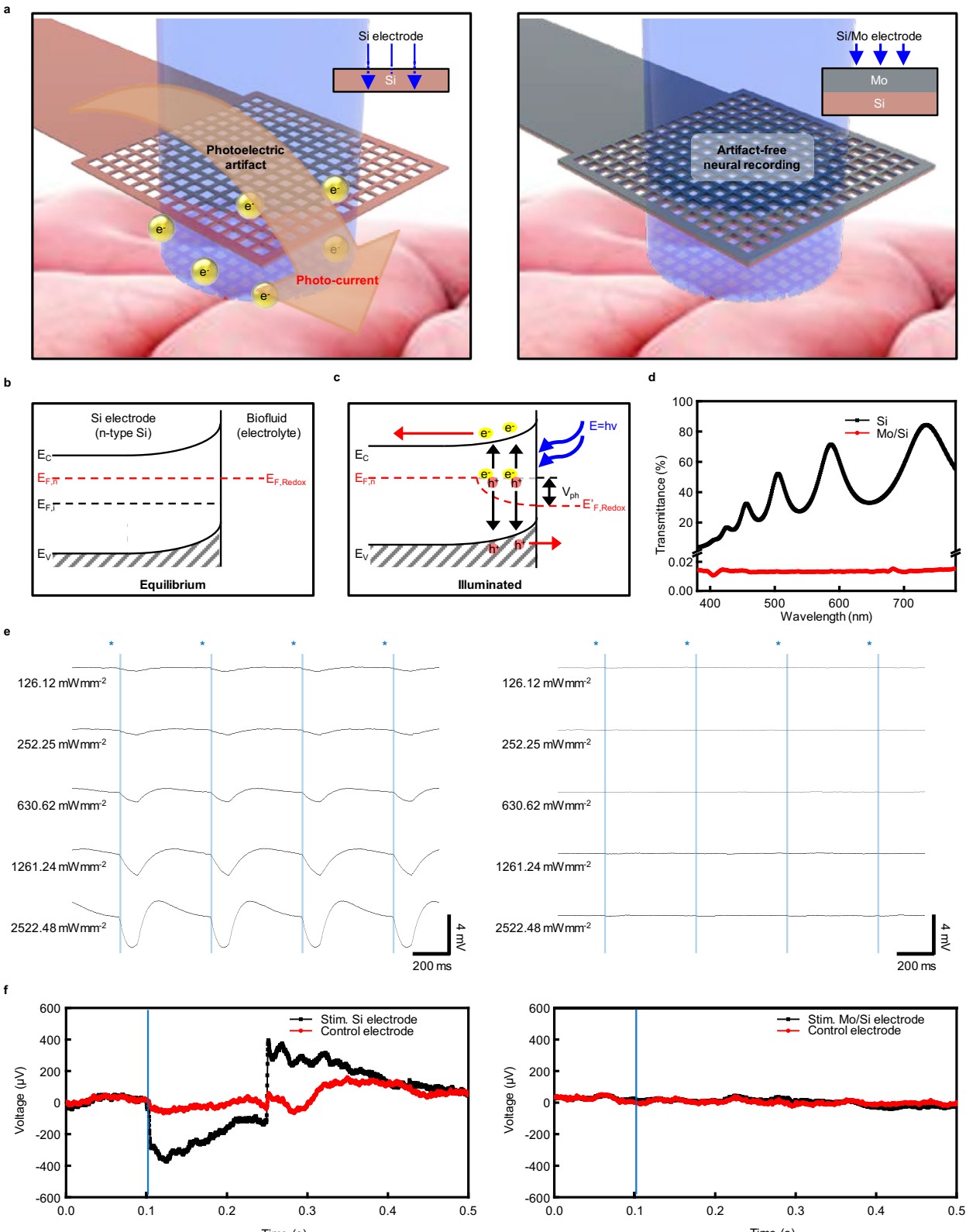

**Fig. 3 | In vitro and in vivo photoinduced artifact evaluations of the biodegradable Si and Mo/Si electrode arrays. a** Schematic illustration of the existing bioresorbable Si electrode and the biodegradable Mo/Si bilayer electrode upon illumination with a 460 nm laser. Energy band diagrams at the n-Si electrode and biofluid interface under **b** equilibrium and **c** illuminated conditions. The yellow balls indicate electrons and red balls indicate holes. **d** Visible light transmittance spectra of 300 nm-thick Si and Mo nanomembranes. **e** Photoinduced artifact evaluation for monolayer Si and bilayer Mo/Si nanomembrane electrodes performed on biomimetic hydrogels by illuminating with laser pulses of varying intensity. **f** ECoG recordings for monolayer Si and bilayer Mo/Si nanomembrane electrode arrays performed on the cerebral cortex of a wild-type mouse with blue laser pulse stimulation.

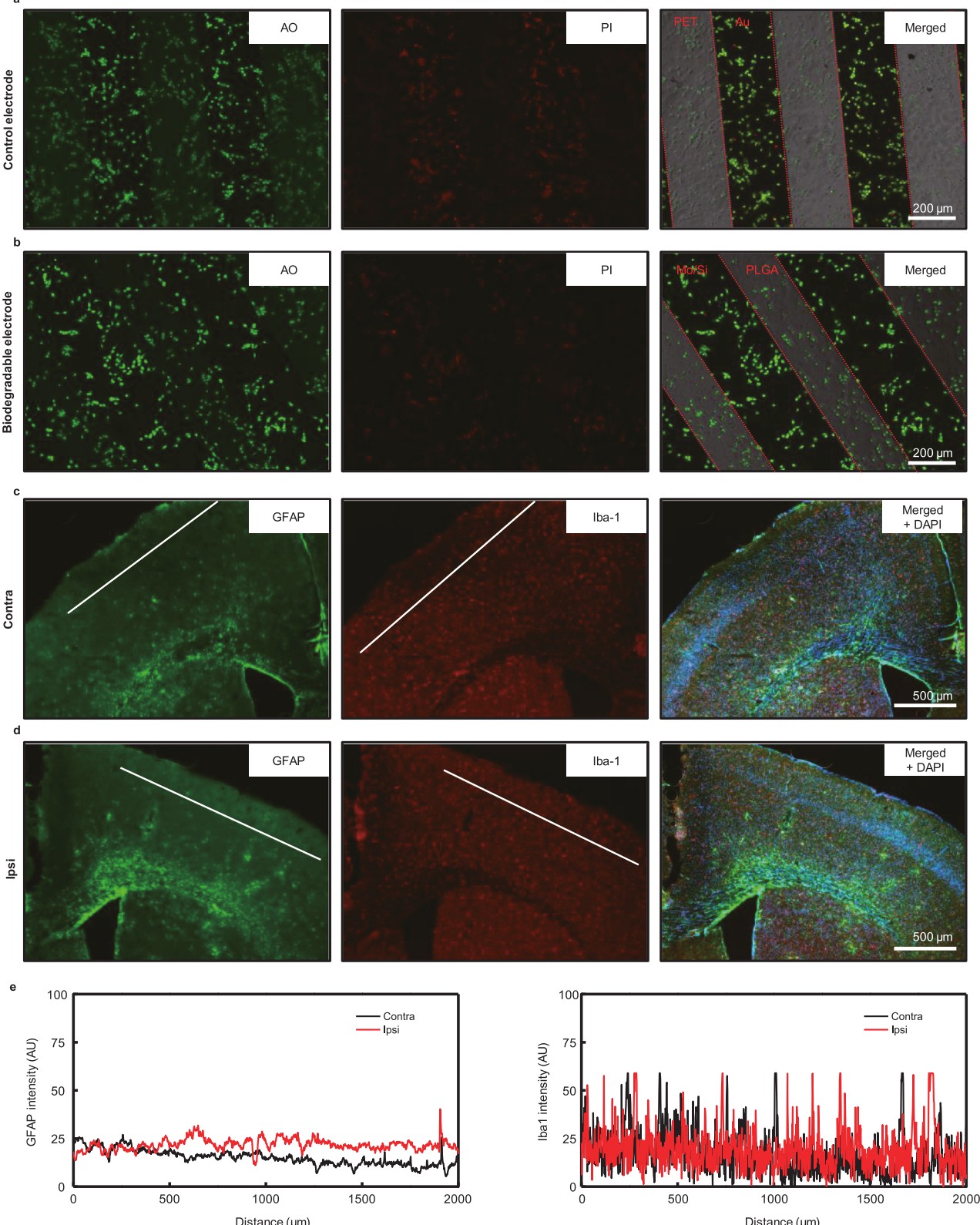

**Fig. 4 | Cell viability tests and immunohistochemistry analysis.** Representative fluorescence microscopy images of live/dead assays of cultured neuronal cells on day 3, stained with acridine orange (AO) (green) and propidium iodide (PI) (red), displayed on **a** control and **b** bioresorbable electrode arrays (left: live cells, center: dead cells, right: merged). Double labeling for astrocytic marker GFAP (green) and microglia marker Iba-1 (red) **c** and **d**, No significant difference of GFAP and Iba-1 intensity in both contralateral and ipsilateral hemispheres. **a-d,** Each experiment was repeated at least three times with similar results. **e** Comparison of immune response by GFAP and Iba-1 intensity (AU = arbitrary units). Representative white lines across superficial cortical layers quantified glial activity. Cell nuclei are visualized with a DAPI stain (blue) (scale bars = 200 µm for **a** and **b**; 500 µm for **c** and **d**).

electrodes on PLGA substrate. To assess the cytotoxicity, neuron viability assay, and images were obtained under fluorescence microscopy. Supplemental Fig. S29 displays representative images of the morphologies of cultured neurons on both the control and biodegradable electrode arrays, revealing a substantial proportion of live, healthy cells within each population. Additionally, cell viability was quantified, indicating that the biodegradable electrode array ensures high cell viability. Furthermore, immune responses were examined through immunohistochemistry. Figure 4c and d display the results following the procedures outlined in the method section. Double immunostaining for astrocytes (GFAP) and microglia (Iba-1), in Fig. 4c and d, reveals glial cell activation 8 weeks post-implantation. In bioresorbable devices, moderate subdural gliosis occurs at the implantation site, when compared to the control contralateral hemisphere. As shown in Fig. 4e, notably, there was no significant difference in immune responses between the region of device placement and the control contralateral hemisphere. These results indicate the high biocompatibility of the materials used in the device, ensuring its stable performance.

### In vivo ECoG recording and optogenetic stimulation

An in vivo ECoG recording test including the hybrid bioresorbable system was performed on the cerebral cortex of an adult Thy-1: ChR2 transgenic mouse, as depicted in Fig. 5a. Under urethane anesthesia, the mouse was secured in a stereotaxic apparatus, and a craniotomy was performed on the left hemisphere to expose a $5 \times 8\,mm^2$ area of the cerebral cortex. As a control, a conventional gold electrode was symmetrically placed in the right hemisphere. Figure 5b displays similar recorded signals for the spontaneous neural activities and K-complex measured under urethane anesthesia, from one of the channels of the biodegradable Mo/Si electrode and the control Au electrode. To record seizure-like electrophysiological signals, pilocarpine was administered via intraperitoneal injection to the mouse (Supplemental Fig. S30). Pilocarpine, which acts by mimicking the neurotransmitter acetylcholine, functions as a cholinergic agonist and activates muscarinic receptors within the nervous system[60]. Seizure-like spiking activities induced by pilocarpine were recorded by the representative electrodes from both the bioresorbable and control electrode arrays, as presented in Fig. 5c and Supplemental Fig. S31. The spectrogram analysis reveals that the urethane-anesthesia activity appears similar to typical sleep spindle activity, and the seizure-like spiking activity shows pilocarpine-evoked LFPs recorded between 0 and 25 Hz (Supplemental Fig. S32)[61–63]. These results confirm that the Mo/Si biodegradable electrodes exhibit the ability to reliably record ECoG signals, similar to the typical Au metal electrodes.

Next, a chronic in vivo optogenetics experiment was conducted involving simultaneous optical stimulation and recording of evoked LFPs in the cerebral cortex of the adult Thy-1: ChR2 transgenic mice (Supplemental Fig. S33). Similar to the in vivo ECoG recording, a biodegradable hybrid system equipped with a head-stage, ACF cable, and optical cannula was placed on an exposed $5 \times 8\,mm^2$ area of the cerebral cortex through craniotomy (Supplemental Figs. S34 and S35). Details are explained in the "Methods" section. Figure 5d shows photographs of the biodegradable hybrid system implanted in the cerebral cortex of a Thy-1: ChR2 transgenic mouse, highlighting the delivery of 460 nm blue light through the waveguide for targeted optogenetic stimulation. Upon exposure to 460 nm pulses transmitted from a waveguide (input power: 24 mW, stimulation intensity: 63.08 mW mm$^{-2}$), ChR2 ion channels within neurons in a specific target region of the cerebral cortex of the transgenic mouse are depolarized without overheating the adjacent local tissue, leading to the generation of action potentials and subsequent activation of neurons (Supplemental Fig. S36)[64]. Figure 5e shows the spectrogram of evoked LFP power spectral density in the frequency range from 0 to 300 Hz. The dotted lines highlight the time point when the LFP activities are evoked

by optical stimulation. Compared to the state without external stimulation, the evoked local field potentials induced by optical stimulation with 460 nm pulses (intensity: 63.08 mW mm$^{-2}$, duration: 30 ms, frequency: 2 Hz) were particularly strong at frequencies below 30 Hz and exhibited periodicity, extending up to 300 Hz. Figure 5f display the evoked LFPs recorded from each electrode channel of the bioresorbable hybrid system, induced by 460 nm pulses (intensity: 63.08 mW mm$^{-2}$, duration: 30 ms, frequency: 2 Hz stimulation). On day 0, in contrast to the control group of wild-type mice, the peak of the optogenetically induced evoked LFPs in the cerebral cortex of Thy-1: ChR2 transgenic mice occurred ~15 ms after initiation of stimulation, with magnitudes ranging from 200 to 250 µV (Supplemental Figs. S37 and S38). Additionally, as the stimulation intensity increased from 15.77, 31.54, to 63.08 mW mm$^{-2}$, evoked LFP peaks of 100, 160, and 230 µV, respectively, were observed (Supplemental Fig. S39). The device operates in a stable manner without any significant change in performance, recording similar evoked LFPs in the same stimulation condition for 14 days following the initial implantation. Starting with malfunctions of the device on day 20, several channels exhibit signs of malfunction by day 21, due to the occurrence of an open-circuit state of the electrodes caused by bioresorbtion[34]. And the remaining operational channel exhibits a decrease in light transmission efficiency due to the in vivo degradation and erosion of the PLGA waveguide, leading to a reduction in the peak amplitude of the recorded LFPs to below 150 µV attributed to the diminished light intensity for stimulation[65]. Consequently, all channels had failed by the 28 days post-implantation. Additionally, the dissolution process of the implanted device was monitored using computed x-ray tomography. The implanted device was traceable on day 21, after which it completely dissolved by day 50 (Supplemental Fig. S40). Collectively, these results suggest that the fully implanted bioresorbable hybrid device operated transiently while maintaining its performance, successfully recording evoked LFPs with minimal artifacts and simultaneously transmitting light from a laser source for optogenetic stimulation, ultimately dissolving and disappearing in vivo.

## Discussion

In summary, this study presents a bioresorbable hybrid optoelectronic system that seamlessly combines optogenetic stimulation and electrophysiological recording in a single unit. The waveguide and electrode structural design efficiently optimizes light transmission, allowing selective and accurate stimulation of target cerebral cortex regions. This hybrid device, which is composed of a biodegradable bilayer Mo/Si electrode array integrated with a soft and flexible PLGA waveguide, successfully eliminates photo-induced artifacts caused by the optoelectronic integration of conventional recording electrodes with optical devices. Moreover, the bioresorbable hybrid system was tested in vitro and in vivo, demonstrating high biocompatibility indicating its suitability for implantation without significant biological impediment. Furthermore, chronic implant in vivo experiments on the cerebral cortex of adult Thy-1: ChR2 transgenic mice confirmed the ability of the device to reliably capture both physiological and pathological ECoG, as well as evoked LFPs during optogenetic stimulation for a period exceeding two weeks, after which it completely biodegrades and disappears within the body.

As a result, the bioresorbable hybrid optoelectronic device is thus promising in the field of neuroscience and neurotechnology, providing researchers with a robust tool for investigating neural circuits and understanding complex interactions between optogenetic manipulations and neuronal responses. By enabling precise, targeted optical stimulation and real-time recording of neural activities, this optoelectronic device is anticipated to contribute significantly to the theragnosis of various neurological disorders necessitating temporal, focused intervention, such as epilepsy. Additionally, its biodegradable nature mitigates the risk of long-term complications inherent to

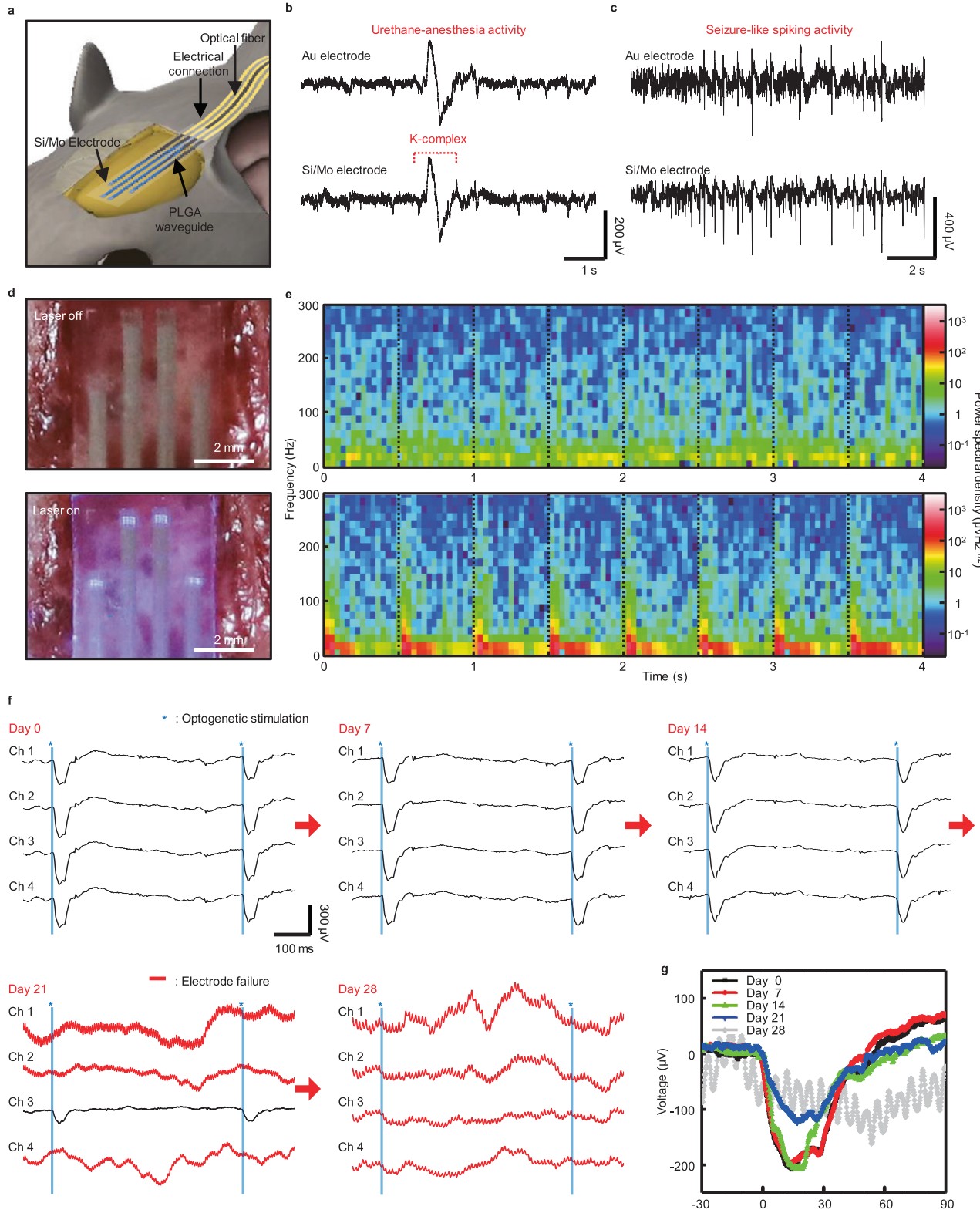

**Fig. 5 | In vivo ECoG recording and optogenetic stimulation in the Thy-1: ChR2 mouse involving a fully bioresorbable hybrid neural implant system.**
**a** Schematic illustration of the biodegradable hybrid system implanted in the cerebral cortex of the Thy-1: ChR2 mouse. **b** Spontaneous and **c** spiking activities captured by the bioresorbable and control electrodes. **d** Photographs of the implanted biodegradable device constructed with PLGA waveguide and 4-channel electrode array implanted at the cerebral cortex when the laser is off (top) and on (bottom). **e** Spectrogram of spontaneous activities (top) and evoked LFPs (bottom) power spectral density. Representative ECoG signals of Thy-1: ChR2 mouse from chronic recording experiments, captured by a fully implanted bioresorbable hybrid device for days 0, 7, 14, and 21. **f** ECoG including evoked LFPs recorded by 4-channel electrode array for pulsed photostimulation (intensity: 63.08 mWmm⁻², duration: 30 ms, frequency: 2 Hz) on days 0, 7, 14, 21, and 28. The blue line indicates the moment of stimulation. By day 21, the device exhibits a decline or loss of function. **g** Sorted optogenetic LFP responses recorded from the bioresorbable electrode array during chronic implantation. Black, blue, green, red, and gray lines correspond to the ECoG signals recorded on days 0, 7, 14, 21, and 28, respectively.

traditional implants and simplifies the device extraction process, which is expected to bring substantial improvements to clinical methodologies. Moving forward, synergistic advancements across various engineering domains will enable enhanced cost-effectiveness in the manufacturing process, precise control of biodegradation kinetics, scaling up to high-density and large-area formats, and integration with wireless systems, collectively making it a more favorable choice for future clinical applications.

## Methods

### PLGA waveguide fabrication

Fabrication of the poly(lactic-co-glycolic acid)(PLGA) waveguide starts with the fabrication of the master mold. On the prepared glass substrate, polyimide (PI, ~4 μm, PI-2545, HD MicroSystems) is spun coated, baked (150 °C for 10 min, 210 °C for 180 min), and Cu (~300 nm) is deposited by sputtering (KVS-T8860, Korea Vacuum). After patterning the waveguide pattern through photolithography, PI and Cu are etched, sequentially. And spin-casting SU-8 100 (~100 μm) on a glass substrate. Photoexposure of the SU-8 100 (Microchem) layer after soft baking (65 °C for 20 min, 95 °C for 90 min) was carried out for patterning waveguides in a state where the glass substrate was turned over and fixed zig to tilt the master mold at 60° (Supplemental Fig. S13). After post-exposure baking (65 °C for 1 min, 95 °C for 15 min) and development, the fabricated master mold was fixed on a petri dish (diameter: 6 inch) and 5:1 polydimethylsiloxane (PDMS, Sylgard 187, Dow Corning) was poured to obtain a PDMS mold. The completely cured PDMS mold was separated from the master mold and petri dish before being fixed on another glass substrate with Kwik-sil (World Precision Instruments) to prevent deformation of the mold. A fabricated PLGA-chloroform solution (5%w/v) (PLA:PGA = 75:25, Mw = 85,000 g/Mol, IV = 0.85 dL/g, Rimless Industry Co.) was poured over the PDMS mold and dried at room temperature to form the PLGA waveguide film. The fabricated dry film was not separated from the PDMS mold before electrode array transfer printing.

### Mo/Si electrode array fabrication

Fabrication of the electrode array started with solid-state phosphorus doping (PH-1000N Source, Saint Gobain) at 950 °C for 15 min of a silicon-on-insulator (SOI, top Si ~300 nm, SOITEC) wafer. We separated the Si nanomembrane (SiNM) on the PDMS slab by wet etching the buried oxide layer of the SOI wafer with concentrated HF (Sigma-Aldrich) and transfer-printed on a Si wafer substrate upon which poly(methylmethacrylate) (PMMA, ~1 μm, A8, Microchem) and polyimide (PI, ~150 nm) were spin-cast sequentially. A Mo layer (~300 nm) was deposited through e-beam evaporation (Korea Vacuum Tech) and the device pattern was defined through photolithography and reactive ion etching (RIE, Young Vacuum System) for Mo, Si NM patterning. The PI top layer (~150 nm) was spin cast and patterned for accessibility of acetone for PMMA dissolution. The multilayer PI, Si, Mo, and PI were separated from the Si substrate by PMMA dissolution in acetone and retrieved reversely from the Si wafer for bottom PI layer etching by RIE. The PI, Mo, and Si multilayer on the Si wafer was stamped using a PDMS stamp, and the top PI layer was etched by RIE. The remaining Si, Mo multilayer was transferred to another PMMA (~1 μm), PI (~150 nm) spin-cast Si wafer substrate. The order of the above processes was to change the order of the Si and Mo layers. A layer of SiO₂ (~150 nm) was formed through plasma-enhanced chemical vapor deposition (URECA 2000, Jusung Engineering). Buffered oxide etchant(10:1, Sigma-Aldrich) was used to remove the SiO₂ layer from the electrode regions, and a top PI layer (~150 nm) was spin cast and patterned for a neutral mechanical plane.

### Device integration

The fabricated electrode array was separated from the Si wafer substrate by PMMA dissolution in acetone, and the retrieved Si wafer was used for top PI layer etching by RIE. The electrode array was stamped with a PDMS stamp and the bottom PI layer was etched by RIE. The PDMS slab with the electrode array was stamped with the PLGA waveguide film whose surface is lightly swollen by an acetonitrile solvent to remove the PDMS slab after drying. We completed the integration by separating the PLGA waveguide film with the electrode array and PMDS master mold. Then, ACF cables (Elform) were bonded to the via regions of the electrode array interconnects for connection to an external data acquisition system. Additionally, fiber optic cannula (Ø1.25 × 6.4 mm ceramic ferrule, Ø105 μm core, 0.22 NA, l = 2 mm, Thorlabs) were precisely aligned to the entrance of the PLGA waveguides on the customized head-stage made by 3D printing and bonded using Norland Optical Adhesive 76 (NOA 76, Norland Products).

### Refractive index measurement

Prepared each PLGA-chloroform solution (5 %w/v) (PLA:PGA = 75:25, Mw = 85,000 g/Mol, IV = 0.85 and PLA:PGA = 50:50, Mw = 90,000 g/Mol, IV = 0.90 dL/g, Rimless Industry Co.) was poured onto a Teflon dish (diameter: 2 inch) and dried at room temperature to create 100 μm thick PLGA films. Films cut to 8 × 8 mm were loaded into the prism coupler (2010/M, METRICON), and the refractive index was measured.

### Electrochemical impedance spectroscopy

Electrochemical impedance spectroscopy (EIS) measurements were obtained using a Gamry Reference 600+ potentiostat (Gamry Instruments). Impedance values were obtained using a three-electrode configuration consisting of an Ag/AgCl reference electrode, a large-surface-area Pt counter electrode, and a working electrode specific to the experimental setup. The frequency was swept from 1 Hz to 10 kHz for devices immersed in PBS (pH 7.4) with an AC measurement voltage of 10 mV. The oscillation amplitude was fixed at 10 mV, and the number of points per decade was set to 10.

### Tensile test of the PLGA waveguide substrate

The PLGA waveguide substrate was subjected to a tensile test at room temperature using a z-axis electric pull tester (KMX-E1000N, MAS) operating at a controlled speed of 8 mm/min. Simultaneously, the tensile force generated during stretching was measured using a force gauge (DTG-10, Digitech). The test specimen utilized for measurement possessed a 5 mm width, 10 mm length, and a thickness of 100 μm.

### Bending test of the device

Bending tests were performed on devices wrapped around a cylindrical glass rod with a radius of 4 mm. The electrochemical impedances of the devices were measured before and after bending as well as after each bending cycle using a Gamry Reference 600+ potentiostat. The same experimental protocols were adopted as those of the EIS measurements (sweep frequency: 1 Hz–10 kHz, AC measurement voltage: 10 mV).

### Accelerated degradation tests of the device

The devices were immersed in a gently stirred (1 Hz) buffer solution (Samchun Pure Chemicals, pH 9.87 at 37 °C). To prevent evaporation of the solution, the container was fully sealed with a PDMS lid. Photographs were captured at 7-day intervals to monitor the progress.

### Visible light transmittance measurement

The grid-patterned Mo/Si (thickness 600 nm) and Mo (thickness 300 nm), Si (thickness 300 nm) NMs transferred onto glass slides were measured using a UV/Vis spectrophotometer (V-650, JASCO). The baseline was corrected using a PI film (150 nm) sample on a glass slide. Each sample was scanned three times in the visible light range between 380 and 780 nm to obtain the measurements.

## Ray-tracing simulation

To perform a comprehensive 3D ray-tracing simulation based on the Monte-Carlo method, commercial software called OpticStudio 16.0 by ZEMAX, Inc. was utilized. The simulation employed a monochromatic point light source with a cone angle of 19.5°, inducing TIR in the waveguide (Supplemental Fig. S8), located at the entrance of the waveguide structure. The refractive index of the waveguide was set to 1.47, and for increased accuracy, the refractive indices of the Si and metal were obtained from measured results[66,67]. The simulation used $1 \times 10^7$ rays to obtain stable calculation results, and a rectangular detector with dimensions of $2500 \times 500$ pixels in the $x$ and $y$ directions was employed to record the light propagation from the entrance to the tip of the waveguide. 2D light distributions were reconstructed using MATLAB software by Mathworks Inc.

Path analysis in OpticStudio is conducted to specify absorption loss at the anti-artifact layer. The result of the path analysis is graphically represented using Origin 2022 by OriginLab Corporation.

## Noise filtering and quantitative comparing in measurement and simulation

For the measurement data, the first step was converting the raw RGB image files into grayscale data. Next, all noise that was <5 % of the maximum value was removed by assigning a value of zero. The remaining data was summed to obtain the total light amount in the noise-removed image. The number of active pixels was counted and the total intensity value was divided by the number of active pixels to obtain the mean intensity per pixel.

In contrast, for the simulation data, RGB to grayscale conversion was not required, as the data was already in the form of intensity with units of $W/m^2$. Noise <5 % of the maximum value was set to zero, and the remaining values were summed. The number of active pixels was then counted, and the total intensity value was divided by the number of active pixels to obtain the mean intensity per pixel.

The values shown in Supplemental Fig. S8 were obtained using the above-described process. To investigate whether the average intensity per pixel changes with the angle of the waveguide tip, normalization was performed by comparing the results obtained at a 50° angle to those obtained at a 90° angle.

## In vitro photo-induced artifact evaluation

All experiments were conducted in a light-shielded darkroom. The measuring and reference electrodes were placed in 0.7 % agarose gel (1× TAE (Tris–Acetate–EDTA) Buffer, E&S Bio Electronics Company). Pulses of 460 nm wavelength light (105 μm core, 0.22 NA SMA905 to Ø1.25 mm Ferrule Patch Cable, Thorlabs) (duration: 100 ms, 2 Hz) generated from a blue diode laser (MDL-lll-460 100 mW, CNI laser) were irradiated on the backside of the electrode. The light intensity emitted from the end of the optical fiber was measured using a power meter (1936-R, Newport Inc., Irvine, CA, USA), and the light power values irradiated from the measured optical fiber were 1, 5, 10, and 20 mW. Signals were then recorded using an Intan system (RHD 2000 EVALUATION BOARD Version 1.0, Intan Technologies).

## Animal preparation and surgery

All the animal work was performed under the study protocol KIST-IACUC-2022-155, as approved by the Institutional Animal Care and Use Committee of the Korea Institute of Science and Technology (KIST). And the animals were kept on a 12-h light–dark cycle with controlled temperature ($21 \pm 1$ °C) and humidity ($50 \pm 10$ %) and had *ad libitum* access to food and water. The experiments utilized adult male wild mice (C57BL/6) and Thy-1: ChR2 transgenic mice (C57BL/6) that were 8–10 weeks old and weighed 25-30 g at the time of bioresorbable device testing. The mice were bred in a controlled animal facility with appropriate environmental conditions. The mice were anesthetized with 0.5 % urethane (400 mg kg$^{-1}$, intraperitoneal injection) or

maintained under 1–1.5 % isoflurane and fixed to the stereotaxic instrument (Model 940, David Kopf Instruments, Tujunga, CA) for surgery. To implant the bioresorbable device into the primary somatosensory cortex (AP: −1.5 mm, ML: −3 −−0.5 mm from bregma), we marked the target area based on the mouse atlas of Paxinos and Franklin. The skull and the dura mater were gently removed with a size of $5 \times 8$ mm$^2$ around the target areas. Then, the device was placed in the designated target area and was used to record in vivo neural signals. For chronic in vivo recording, we put dura-gel (Cambridge Neurotech) on the exposed dura, and additional screws were secured in the skull for anchoring. The skull and device were then covered with dental cement.

## In vivo photo-induced artifact evaluation

The experiments utilized adult male wild mice (C57BL/6) that were 8–10 weeks old and weighed 25–30 g at the time of bioresorbable device testing. The mice were bred in a controlled animal facility with appropriate environmental conditions. The mice were anesthetized with 0.5 % urethane (400 mg kg$^{-1}$, intraperitoneal injection) and fixed to the stereotaxic instrument (Model 940, David Kopf Instruments, Tujunga, CA) for surgery. To implant the bioresorbable device into the primary somatosensory cortex (AP: −1.5 mm, ML: −3 - −0.5 mm from bregma), we marked the target area based on the mouse atlas of Paxinos and Franklin. The skull and the dura mater were gently removed with a size of $5 \times 8$ mm2 around the target areas. Then, the device was placed in the designated target area and was used to record in vivo neural signals. Pulses of 460 nm wavelength light (105 μm core, 0.22 NA SMA905 to Ø1.25 mm Ferrule Patch Cable, Thorlabs) (duration: 100 ms, 2 Hz) generated from a blue diode laser (MDL-lll-460 100 mW, CNI laser) were irradiated on the backside of the electrode. The light intensity emitted from the end of the optical fiber was measured using a power meter (1936-R, Newport Inc.), and the light power values irradiated from the measured optical fiber were 1, 5, 10, and 20 mW. Signals were then recorded using an Intan system (RHD 2000 EVALUATION BOARD Version 1.0, Intan Technologies).

## Cell viability test

Hippocampal neurons were isolated from the hippocampi of Sprague−Dawley rat embryos (E18) using Hank's balanced salt solution (Thermo Scientific), and subjected to enzymatic digestion using carefully optimized trypsin and DNase solution with fire-polished Pasteur pipettes. Dissociated neurons were plated on Au electrodes on a PET substrate and Mo/Si electrodes on a PLGA substrate and maintained in a neurobasal medium containing B27 (Life Technologies), L-glutamine(Life Technologies), and penicillin/streptomycin in 5 % CO$_2$ at 37 °C. At days in vitro 3, the cells were stained with acridine orange/ propidium iodide (PI/AO; 1:100; Aligned Genetics, Anyang, Republic of Korea) for neuron viability assay, and images were obtained under fluorescence microscopy. Live cells were labeled with green fluorescence and dead cells with red fluorescence. The digital color fluorescence microscope (Invitrogen M5000, EVOS™) was used to count the live and dead cells.

## Immunohistochemistry

To evaluate the immune reactions to the biodegradable device in the brain, we examined activated astrocytes using GFAP and identified activated macrophages and microglia using Iba-1 in the coronal section of the somatosensory cortex. In vivo experiments were conducted over a duration of 2–8 weeks to evaluate immune responses. Anesthetized mice were sacrificed through transcardial perfusion of 4 % [w/ v] paraformaldehyde (PFA, Sigma-Aldrich) in 0.1 M PBS. The dissected brain was immersed in a 4 % PFA solution in 0.1 M PBS for 24 h at 4 °C. Subsequently, the fixed brain underwent cryoprotection using 30 % sucrose (w/v) in 0.1 M PBS for 48 h at 4 °C. The cryoprotected brain was then horizontally sectioned into 40 μm-thick slices. These slices,

measuring 40 μm in thickness, were rinsed in 0.1 M PBS and subjected to blocking in a solution comprising 0.1 % Triton X-100(Sigma-Aldrich) (v/v) and 3 % bovine serum albumin (BSA, Sigma-Aldrich) (w/v) in 0.1 M PBS for 1 h at room temperature. Following three 30-minute washes in 0.1 M PBS, the brain slices, which were 40 μm thick, underwent incubation with primary antibodies (Chicken-anti-GFAP, 1:500, AB5541, EMD Millipore Corp.; Rabbit-anti-Iba1, 1:200, 019-19741, Wako) in 0.1 M PBS containing 3 % [w/v] BSA for 24 h at 4 °C. After being washed three times with 0.1 M PBS for 30 minutes, the brain slices, which had been treated with primary antibodies, were subjected to incubation with secondary antibodies (donkey-anti-chicken conjugated Alexa Fluor 488, 1:500, #703-545-155, Jackson; donkey-anti-rabbit conjugated Alexa Fluor 594, 1:500, #711-585-152, Jackson) in 0.1 M PBS containing 3 % [w/v] BSA for 2 h at room temperature. Following three 30-minute washes in 0.1 M PBS, the brain slices were treated with 4′,6-diamidino-2-phenylindole (DAPI; 1:3,000, D1306, Pierce) in 0.1 M PBS containing 3 % BSA for 1 h at room temperature. Lastly, the samples were mounted with a fluorescent mounting medium (S3023, Dako) and dried.

### Confocal imaging and image quantification
The images of these samples were obtained with a confocal microscope (A1, Nikon), and Z stack images in 3-μm steps were conducted. The confocal microscopy settings were kept constant for all samples and controls (laser power, filters, dichroic mirrors, polarization voltage, and scan speed). The exposure parameters were also consistent across all individual images. Any conversion in brightness or contrast was evenly applied to the entire image set. Confocal microscopic images were analyzed using the ImageJ program (NIH). To measure the spatial cross-correlation of GFAP and Iba1 signals, a manual line was drawn across the neocortex at the site of the neural implant application to define the region-of-interest and we conducted a line-plot analysis using the plot profile function in ImageJ, based on previous reports[68,69].

### Drug delivery
For in vivo recording of seizure-like electrophysiological signals, 150 mg of pilocarpine (Sigma-Aldrich) was administered via intraperitoneal injection in the mouse 30 min after the administration of methyl bromide.

### In vivo temperature monitoring
To monitor the operational temperature of the device and verify its thermal stability, the temperature was continuously monitored using a digital thermometer (TX10, Yokogawa) before and after a 5-min stimulation period. Following the same procedure as the in vivo photo-induced artifact evaluation, the device was implanted in the somatosensory area of an adult wild-type mouse with an opened skull. Positioning the thermometer 15 cm away from the device, the temperature was measured.

### In vivo assessment of microarchitecture using micro-computed tomography in a murine model
Mice were anesthetized using isoflurane, and anesthesia depth was monitored throughout the imaging procedure. Once anesthetized, animals were positioned in the Micro-CT scanner (SkyScan 1276, Bruker) with limbs immobilized to reduce motion artifacts. A scout view was obtained to ensure proper positioning, followed by the acquisition of high-resolution 3D images with settings optimized for bone imaging (60 kV, 200 μA, 0.5 mm aluminum filter, isotropic voxel size of 30 μm). Raw Micro-CT data were reconstructed using NRecon software (Bruker) to generate cross-sectional images. CTAn software (Bruker) was employed for quantitative analysis of bone microarchitecture parameters, including bone volume fraction (BV/TV), trabecular thickness (Tb.Th), trabecular separation (Tb.Sp), and trabecular number (Tb.N).

The reconstructed images were then subjected to three-dimensional (3D) volumetric rendering to visualize bone microstructure.

### Electrophysiology and data analyses of in vitro and in vivo experiments
Using the Intan (RHD 2000 EVALUATION BOARD Version 1.0, Intan Technologies) recording system, the neural signals were recorded, amplified, and digitized from the recorded electrical signals through the bioresorbable device (notch filter: 60 Hz, bandpass filter: 1-300 Hz). All data were stored as RHD files. The data were then imported into MATLAB as "double" format for signal loading and processing. All data were in raw form, without any subsequent signal processing.

### Statistics and reproducibility
An appropriate sample size was computed when the study was being designed. Before we started the statistical tests, such as two-tailed Student's $t$-test, and ANOVA, we confirmed that the data we obtained passed the normality test. If not, we carried out the Mann–Whitney test. The electrochemical impedance plots measured for the bending durability tests and grid designs were represented as each mean value of $n = 3$ and 4 samples, while the transmittance plots of the nanomembranes were depicted as the mean of $n = 3$ samples by using Origin Pro 8.10 (Origin Lab) software. Three independent group comparisons of Au and Mo/Si electrodes were analyzed by two-tailed Student's $t$-test or Mann–Whitney test. All data are presented as mean ± SEM. All statistical analyses were performed using GraphPad Prism software (GraphPad Software Inc.). All optical microscope images were obtained by repeating the experiment at least three times to ensure consistent results.

### Reporting summary
Further information on research design is available in the Nature Portfolio Reporting Summary linked to this article.

## Data availability
All data supporting the findings of this study are available within the article and its supplementary files. Any additional requests for information can be directed to, and will be fulfilled by, the corresponding authors. Source data are provided with this paper.

## Code availability
MATLAB codes for Supplemental Figs. S12, S16, and S17 are available in the Zenodo-linked GitHub repository at https://doi.org/10.5281/zenodo.10581398.

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

## Acknowledgements
This work acknowledges the support received from the National Research Foundation of Korea (Grant nos. NRF-2019R1A2C2086085, NRF-2021R1A4A1031437 (K.J.Y.), NRF-2022M3E5E8081196 (I-J.C.), NRF-2021R1C1C2013605, RS-2023-00217312 (G.J.L.), NRF-2019H1A2A1077020 (M.C), and NRF-2021R1A6A3A0108733912 (J.-K.H.)), the KIST Institutional Program (Project No. 2E31603-22-140 (K.J.Y.)) and the Institute for Basic Science (IBS), Center for Cognition and Sociality (Grant No. IBS-R001-D2 (I.-J.C.)). Images in Figs. 1g and 5a from 3D Rat Anatomy Software (www.biosphera.org).

## Author contributions
M.C., J.-K.H., and J.S. wrote the manuscript with input from all co-authors. M.C., J.-K.H., J.S., J.J.K. J.R.R, K.H., Kanghwan K., E.-B.H., and M.-H.N. designed experiments, analyzed the data, and prepared figures. M.C., J.S., I.S.M., M.S., S.L., T.S.K., Kyubeen K., and Kyowon K. performed device fabrication. M.C., J.S., K.H., and J.K. performed measurements on the properties of materials and devices. J.J.K., Y.M.S., and G.J.L. performed the simulation of the devices. M.C., J.-K.H., and J.S. performed the in vitro and in vivo experiments. J.R.R, E.-B.H. M.-H.N. performed cell viability tests and immunohistochemistry analysis. G.J.L., I.-J.C., and K.J.Y. supervised the project.

## Competing interests
The authors declare no competing interests.

## Additional information

[1]Functional Bio-integrated Electronics and Energy Management Lab, School of Electrical and Electronic Engineering, Yonsei University, 50 Yonsei-ro, Seodaemun-gu, Seoul 03722, Republic of Korea. [2]Brain Science Institute, Korea Institute of Science and Technology, 5. Hwarang-ro 14-gil, Seongbuk-gu, Seoul 02792, Republic of Korea. [3]Department of Electronics Engineering, Pusan National University, 2, Busandaehak-ro 63beon-gil, Geumjeong-gu, Busan 46241, Republic of Korea. [4]Department of Anatomy, College of Medicine, Korea University, 17-gil Koryodae-ro, Seongbuk-gu, Seoul 02841, Republic of

Korea. [5]School of Mechanical Engineering, Yonsei University, 50 Yonsei-ro, Seodaemun-gu, Seoul 03722, Republic of Korea. [6]Center for Brain Function, Korea Institute of Science and Technology 5, Hwarang-ro 14-gil, Seongbuk-gu, Seoul 02792, Republic of Korea. [7]School of Electrical Engineering and Computer Science (EECS), Gwangju Institute of Science and Technology (GIST), Gwangju 61005, Republic of Korea. [8]Department of Convergence Medicine, College of Medicine, Korea University, 17-gil Koryodae-ro, Seongbuk-gu, Seoul 02841, Republic of Korea. [9]Department of Anatomy, College of Medicine, Korea University, 7-gil Koryodae-ro, Seongbuk-gu, Seoul 02841, Republic of Korea. [10]Department of Electrical and Electronic Engineering, YU-Korea Institute of Science and Technology (KIST) Institute, Yonsei University, 50, Yonsei-ro, Seodaemun-gu, Seoul 03722, Korea. [11]These authors contributed equally: Myeongki Cho, Jeong-Kyu Han, Jungmin Suh. ✉e-mail: gjlee0414@pusan.ac.kr; ijcho@korea.ac.kr; kijunyu@yonsei.ac.kr

