## [Peer Review File · Nature Communications]

REVIEWER COMMENTS

Reviewer #1 (Remarks to the Author):

The manuscript titled “A fully Bioresorbable Hybrid Opto-Electronic Neural Implant System for Simultaneous Electrophysiological Recording and Optogenetic Stimulation” integrate biodegradable waveguide and neural interface for simultaneous stimulation and recording. I acknowledge that this is a compelling and significant subject for research, given that the presented system exhibits both stimulation and recording multifunctionality, as well as engineering techniques for minimizing light artifacts, all achieved through the use of biodegradable materials. Overall, the authors present a well written paper with interesting ideas to develop such ideas. I recommend the paper to be published after following minor revisions.

* The literature review is fairly detailed but a few more area could be addressed. For example, target applications that are appropriate for the use of biodegradable neural interfaces, rather than those that function chronically, should be considered. It may be more suitable to utilize chronically functioning neural interfaces for conditions like Parkinson’s disease, although such disease is mentioned in the conclusion.

* The conclusions and perspective part can be elaborated more. Potential challenges and enhancements for future devices could also be addressed. One such challenge could be activating biodegradability after a sustained period of stable functionality, which presents difficulties that must be resolved going forward.

* The waveguide should deliver enough light power to the target stimulating area, and the author uses the 20 mW of light power source for the stimulation. The efficiency of light delivery should be to elaborate the optical functionality of such PGLA waveguide.

* Lifetime of current device structure is 2 months, but the author should provide further clarification on the duration of reliable operation in the body, particularly since the material used is biodegradable.

* Figure 5f is not very intuitive. The labeling of evoked LFP in the spectrogram should be considered.

Reviewer #2 (Remarks to the Author):

This study presents a neural interface focused on the development of a bioresorbable, flexible hybrid opto-electronic system for simultaneous electrophysiological recording and optical stimulation. This proposed device addresses the limitations of current bioabsorbable neural implants, which typically perform only single functions. The system is composed of biodegradable PLGA and demonstrates biocompatibility, enabling direct optical and electrical interfacing with the curved cerebral cortex surface. The device has been optimized to minimize light transmission losses and photoelectric artifact interference, and its effectiveness has been demonstrated through in vivo experiments with transgenic mice. The authors highlight the potential of this novel hybrid neural implant system for monitoring neural activities and providing therapeutic interventions in various biomedical applications. The primary critiques of this study emphasize the need for a more comprehensive evaluation and characterization of the as-fabricated devices, both pre- and post-implantation. While the research presents valuable contributions to the field of neural interface development, the limitations and concerns raised suggest that further investigation is necessary to establish this work as a significant advancement in the field.

Major questions:

1. The authors claimed that the entire device is bioresorbable, and the lifetime of the device can be adjusted by controlling the materials parameters such as thickness, doping concentration of Si, and the composition ratio of PLGA. Also, an insulating layer composed of silicon dioxide (SiO₂) was used. However, questions remain regarding the integrity and performance of the encapsulation, and the stability of the entire device, particularly in the context of potential current leakage that may arise as the device degrades over time. To address these concerns, it is essential to provide a comprehensive analysis of the electrochemical properties and electrical insulation under various physiological conditions over the degradation.
2. Connected with the previous comment, why was SiO₂ employed for the insulation and what are the merits of SiO₂ over other biodegradable insulating polymers? It would be valuable to explore alternative biocompatible and bioresorbable insulating materials that could potentially mitigate the risk of current leakage and provide enhanced encapsulation performance. Comprehensive comparative studies on the durability and biocompatibility of various insulating materials could potentially identify alternatives to SiO₂ for bioresorbable device applications.
3. The authors demonstrated the achievement of total internal reflection within the device by establishing a refractive index contrast and presenting ray tracing simulation results based on the refractive index values. However, given different fractions of polymers exhibit different refractive indices, the precise characterization of the refractive index for the specific PLGA formulation used in this study remains unclear and warrants further elaboration. The authors should consider conducting experimental

measurements of the refractive index of the PLGA formulation with different concentrations, detailing the methodology and results in the manuscript.

4. Following the previous comment, the authors presented the transmittance measurements of Mo/Si, Mo, and Si nanomembranes in the visible light range. However, the absorbance/transmittance and autofluorescence properties of the PLGA waveguide were not discussed. Furthermore, the molecular weight of PLGA and its lactic-to-glycolic acid ratio was not mentioned in the methods section, which is a crucial detail, as variations in these parameters can influence the autofluorescence of the polymer. Additionally, the fabrication process involves soft baking above 95°C, which may contribute to increased autofluorescence in the material. It is essential to provide a comprehensive analysis of the absorbance/transmittance and autofluorescence properties of the PLGA waveguide to fully evaluate the device's performance.

5. Bending tests were performed on devices wrapped around a cylindrical glass rod to demonstrate the flexibility and biocompatibility of devices. But it is not enough to evaluate the performance of devices after implantation since the mechanical properties (i.e., young's modulus and elongation) are unknown in this study. Mechanical testing of the devices (i.e., tensile tests) should be included in this study. Additionally, the degradation of device in vivo was not thoroughly examined. Does the device cause any immune response or scar formation during implantation and degradation at the implantation site? A comprehensive assessment should be carried out to determine whether the device elicits any immune responses or scar formation during implantation and degradation at the implantation site. This information is important to neuroscientists and other researchers who may be interested in utilizing these devices.

6. The authors reported optogenetics powers in the range of 73.15 – 1463.04 mW/mm², it is helpful for evaluating the light-induced artifacts with such large powers, but the optogenetic stimulation threshold is way lower than 73.15 mW/mm². Such elevated power levels may inadvertently activate heat-sensitive channels in addition to ChR2, potentially leading to overheating of brain tissue and confounding the experimental results. The authors should measure and report the temperature at both the output of the devices and the implantation sites during optogenetic stimulation to ensure that tissue overheating is not a confounding factor in the study.

7. How to connect the device to the external light source?

8. In figure 5b-c, if the authors claim that it is seizure-like spiking activity, please provide single-spike sorting.

Minor comment:

1. Please double confirm the optogenetic stimulation power listed in the main text (line 358, 367, 371, and 375), the units are inconsistent with the Supplementary Figure S12.
2. Based on the requirements of data presentation by the nature portfolio, please show individual points for the histograms when $n \leq 10$ (Figure 4d).
3. Please normalize the optogenetics power by the area, this may confuse the audience (line 429 and 439).
4. Please report the number of mice used for all in vivo experiments: (i) total number of mice utilized in the study. (ii) number of mice in each experimental group, along with any control groups. (iii) any instances of mice being excluded from the analysis, and the reasons for their exclusion. (iv) information on the randomization and blinding procedures employed, if applicable.
5. In the Conclusion: (i) please include the limitations or future improvements of the as-fabricated bioresorbable devices since they require the sophisticated photolithography processing which is costly and time consuming. (ii) the throughput of the devices is unknown, please discuss the possibility in scaling up these bioresorbable devices since ECoG arrays have been augmented to about 1000 channels by other researchers.

Reviewer #3 (Remarks to the Author):

The paper proposes a resorbable flexible opto-electronic system for simultaneous electrophysiological recording and optical stimulation. In vivo recording is achieved with photo-stimulation on transgenic mice. The paper should have broad interests in the community of neuroscience and neuroengineering. To improve the paper, I have the following comments:

Major issues:

1. Fig. 3a, the authors propose a Mo coated Si electrode design to eliminate the artefacts. At first glance, this seems feasible. However, the authors should note that this design can only block the illumination on the top surface of Si, while a considerable amount of photons can be backscattered by the tissue and

enter the bottom side of Si, inevitably creating artefacts. Besides measurements, optical simulations should be performed to compare the number of photons captured by the Si film with and without Mo coatings.

2. Fig. 3e (left, 20 mW) and Fig. 5e, the shapes of these recorded signals are very much alike, which makes one doubt that the results in Fig. 5e are artefacts, just as those in Fig. 3e (left, 20 mW). These possible artefacts can be ascribed to the reason mentioned in Comment 1. At least two additional in vivo experiments should be performed: (1) Do the experiments in Fig. 5d with wild-type mice; (2) In Fig. 5d, only illuminate one or two waveguide channels, and measure the LFP signals from all the four electrode channels.

3. Mo has very poor reflectivity, which is only ~50% at wavelengths of 400-500 nm. There will be 50% loss at a single bounce! Guess how much power will be lost after multiple bounces? The waveguide loss (unit: dB/cm) should be measured and simulated.

Reflectivity of Mo, see: <https://ieeexplore.ieee.org/document/6052234>

4. It is not clear how the in vivo experiments are operated. Shown in Fig. S21, the mouse is head-fixed with the Si/Mo electrodes and the PLGA waveguides mounted on the exposed cortex. This is a setup established for acute test. In such a setup, one cannot understand the motivation of using the waveguide, since light beam can be directly incident on the cortex. The authors should demonstrate the device's utility in animals with skull and scalp covering the brain, as shown in Fig. 1g. Otherwise, there is no point to use the waveguide.

5. In Fig. 5, Thy-1:ChR2 mice are used. Fluorescence images should be provided to illustrate the expression of ChR2 in the targeted brain region.

6. The paper only focuses on examining the acute performance of the neural implant system, both in vitro and in vivo. As the materials gradually degrade, it can substantially impact long-term performance. For instance, the swelling of PLGA can significantly influence optical properties. The dissolution of Si and Mo can introduce recording artifacts and impair stimulation efficiency. Also, chronic degradation analysis should be performed in vivo, to demonstrate the device's disappearance in the mouse brain.

Other issues:

7. Fig. 3b and 3c, what are the differences between n-Si and p-Si in this case? More discussions should be added.

8. Fig. 1i, why do the authors choose a solution with pH = 4.01? The brain environment should be weak base solution with pH = 7.4. To accelerate the degradation, using a solution with pH > 7 will be better.

9. Fig. 2b, how to form the waveguide with facets with different angles? Fabrication details are needed.

10. Fig. 3e uses the unit of mW, while Line 367 uses the unit of mW/cm². These units should be consistent.

11. Fig. 3f, the control electrodes also exhibit different levels of artefacts. Why?

Reviewer #1

General Comment #1: The manuscript titled “A fully Bioresorbable Hybrid Opto-Electronic Neural Implant System for Simultaneous Electrophysiological Recording and Optogenetic Stimulation” integrate biodegradable waveguide and neural interface for simultaneous stimulation and recording. I acknowledge that this is a compelling and significant subject for research, given that the presented system exhibits both stimulation and recording multifunctionality, as well as engineering techniques for minimizing light artifacts, all achieved through the use of biodegradable materials. Overall, the authors present a well written paper with interesting ideas to develop such ideas. I recommend the paper to be published after following minor revisions.

Our response: We thank the reviewer for this positive comment, and the recommendation to publish in *Nature Communications*. We made our revision with pleasure based on the reviewer’s opinion, and all the details of our modifications are indicated in Our response.

Comment #1: The literature review is fairly detailed but a few more area could be addressed. For example, target applications that are appropriate for the use of biodegradable neural interfaces, rather than those that function chronically, should be considered. It may be more suitable to utilize chronically functioning neural interfaces for conditions like Parkinson’s disease, although such disease is mentioned in the conclusion.

Our response: First, we thank the reviewer for the considerate comment. As the reviewer mentioned, when considering diseases that require bioresorbable neural interfaces, attention should be focused on conditions needing temporary monitoring and treatment. The following are examples of acquired brain disorders where bioresorbable neural implants could be applied:

1. Acute Encephalitis: This condition involves acute inflammation of the brain, which may necessitate intensive monitoring over a short period.
2. Traumatic Brain Injury: Used for monitoring the recovery process following brain injuries due to traffic accidents or other impacts to the head.
3. Post-Surgical Monitoring: Employed temporarily to detect and manage complications after specific brain surgeries, such as tumor removal.
4. Acute Brain Hemorrhage: Utilized to monitor the initial recovery process following a brain hemorrhage.

These conditions typically require short-term treatment or monitoring, after which further intervention is unnecessary. In such cases, bioresorbable neural interfaces are particularly useful as they naturally degrade, eliminating the need for additional surgery to remove the device. However, a congenital disease that specifically requires bioresorbable neural implants, similar to the distinctiveness of epilepsy, is challenging. Epilepsy is a relatively common condition with various types and is extensively studied in neuroscientific research, making it a unique case.

However, other congenital diseases may not have such clear-cut requirements for bioresorbable

implants. So, it's difficult to generalize the need for bioresorbable implants to a specific congenital disease as clearly as epilepsy. It's more appropriate to consider the applicability of bioresorbable implants based on specific patient groups or diagnoses. Therefore, in our conclusion, we have cited epilepsy as a representative example while discussing the applicability of bioresorbable implants for diseases that require temporary diagnosis and treatment.

Our modification to the manuscript:

Line 547: By enabling precise, targeted optical stimulation and real-time recording of neural activities, this opto-electronic device is anticipated to contribute significantly to the theragnosis of various neurological disorders necessitating temporal, focused intervention, such as epilepsy.

Comment #2: The conclusions and perspective part can be elaborated more. Potential challenges and enhancements for future devices could also be addressed. One such challenge could be activating biodegradability after a sustained period of stable functionality, which presents difficulties that must be resolved going forward.

Our response: We thank the reviewer for the insightful advice. In the conclusion section, we discussed several specific strategies regarding the potential challenges and improvements of the device. In addition to addressing control over the timing and rate of the device's biodegradation, as the reviewer mentioned, we also examined considerations necessary for future clinical applications such as, cost-effectiveness, the scalability of the device and integration with wireless systems.

Our modification to the manuscript:

Line 550: Additionally, its biodegradable nature mitigates the risk of long-term complications inherent to traditional implants and simplifies the device extraction process, which is anticipated to bring substantial improvements to clinical methodologies. Moving forward, synergistic advancements across various engineering domains will enable enhanced cost-effectiveness in the manufacturing process, precise control of biodegradation kinetics, scaling up to high-density and large-area formats, and integration with wireless systems, collectively making it a more favorable choice for future clinical applications.

Comment #3: The waveguide should deliver enough light power to the target stimulating area, and the author uses the 20 mW of light power source for the stimulation. The efficiency of light delivery should be to elaborate the optical functionality of such PGLA waveguide.

Our response: We appreciate the reviewer for the insightful comment concerning the reflectivity of Mo and the potential for significant power loss in our waveguide design. The waveguide serves as a transmitter that delivers light generated by an external source to the target location. Unlike typical biodegradable self-emitting devices such as micro-LEDs^{R1}, waveguides enable external control of power supply and heat cooling, allowing for the maintenance of high output levels from the external light source to deliver light at the desired intensity. The transmission loss of the optical waveguide can be compensated for by controlling

the output of the external light source. Nevertheless, extremely high transmission losses prevent the device from maintaining its intrinsic functionality. To elucidate the efficiency and feasibility of the proposed system, it is necessary to consider the effectiveness and losses of the waveguide. Therefore, we first conducted additional measurements on the intensity of the light emitted from the tip of the waveguide. The intensity of the light transmitted through the PLGA waveguide was quantified to verify its effectiveness in delivering light for stimulating the target tissue. Furthermore, to enhance the reliability and precision of our device, we have additionally manufactured a headstage that was not previously implemented. The created headstage seamlessly connects the optical cannula with the device. When the output power of the optical cannula is 26.4 mW, the average light intensity emitted from the end of the waveguide to stimulate target tissues 63.06 mWmm^{-2} . This exceeds the previously mentioned reference value, thereby being sufficient to induce adequate stimulation.

Next, to assess the amount of light absorbed by Mo reflective layer throughout the guiding, we first considered a single-channel waveguide system shown in **Fig. S7a**. The proposed waveguide utilizes not just single incident angle but all possible angles for waveguiding, thus we discretized the angles capable of total internal reflection (TIR) and conducted ray tracing simulations. We simulated a single channel PLGA waveguide with light sources satisfying TIR condition, by controlling incident angle from 0° to 19° . In this system, the loss of waveguide is only attributed to Mo absorption because neither light is absorbed by the bottom Si layer nor leaks out of the waveguide. The waveguide losses for discretized angles, including the cumulative sum of each case, are presented in **Fig. S7b**. With larger incident angle, the number of reflections on the Mo layer rises, leading to increased absorption loss. Cumulative loss in the waveguide length of 2cm is -6.72dB meaning a loss per unit length of 3.36dB/cm. Afterward, we performed simulations with the same waveguide utilizing non-discretized fiber source to assess how much the reflection on Mo layer occurs during propagation. **Fig. S7c** shows the result of the number of rays and cumulative power relative to the number of reflections on the Mo layer. The inset illustrates how the number of reflections varies for rays starting at different incident angles. Substantial amount of light reached the destination after undergoing a few reflections on the Mo layer while retaining its power. Considering all light propagating through the single-channel waveguide, the final delivered power is 24.98% of input.

Supplementary Figure S7 | The single-channel waveguide simulation with Mo reflective layer.

(a) Simulation system to measure loss of single-channel waveguide. The waveguide consists of PLGA waveguide, PLGA substrate, Mo reflective layer and Si nanomembrane. The inset below depicts rays which propagate within the waveguide at various angle. (b) The loss of each discretized angle and cumulative loss. A larger incident angle results in great increased losses due to absorption. The cumulative loss is calculated by adding up the power contributions of each angle. (c) The number of rays in relation to the number of reflections on Mo. The inset illustrates how the number of reflections varies for rays starting at different incident angels.

Subsequently, we conducted simulations for the proposed system shown in **Fig. S8a**. Seven power detector, positioned at 1.5mm intervals, measure the transmitted power at each location to calculate propagation loss. Since only two longer channels reach the 7th detector, we extended two shorter channels through dummy waveguide. **Fig. S8b** shows result of guided power measured from detectors. Initially, 20% of light is leaked through the PLGA substrate and the light either exits the system resulting loss or be guided through the substrate. The absorption loss caused by Mo reflective layer does not occur from the starting point of the waveguide but occurs from the point where the waveguide and Mo reflective layer overlap. As the waveguide overlapping with the Mo reflective layer, linear absorption loss is measured at -7.059dB/cm. Note that the loss is less than real value due to additional loss coming from the length compensation. As mentioned earlier in single-channel waveguide where power remains quite substantial, although the absorption by the Mo reflective layer is not negligible, a significant amount of power is still transmitted due to diverse angles where light can propagate within the waveguide. The loss is sufficiently acceptable because our system is designed for optical propagation at distances less than 1cm.

Supplementary Figure S8 | The actual 4-channel waveguide simulation with Mo reflective layer. (a) Simulation system and detector arrangement for loss measurement. The blue segment at the end of waveguide is an extension of short waveguide to compensate the length mismatch. (b) Loss of the compensated waveguide. Absorption losses occur after where the waveguide overlaps with the Mo layer, indicated green area.

[R1] Lu, D., Liu, T. L., Chang, J. K., Peng, D., Zhang, Y., Shin, J., ... & Rogers, J. A. (2019). Transient Light-Emitting Diodes Constructed from Semiconductors and Transparent

Conductors that Biodegrade Under Physiological Conditions. *Advanced Materials*, 31(42), 1902739.

Our modification to the manuscript:

There is a modification in the contents of the sentence.

Line 294: At the bottom side of waveguide, reflection occurs at the interface between the PLGA substrate and electrode array, which minimizes an artifact of Si electrode. Considerable absorption loss is present since reflection from metal inevitably accompanying absorption. Nevertheless, the proposed waveguide delivers a substantial power enough to stimulate neurons (Supplemental Figure S7 and S8). The guided light is transmitted to each electrode and undergoes reflections once again at the inclined waveguide tips.

Line 496: Upon exposure to 460 nm pulses transmitted from a waveguide (input power: 24 mW, stimulation intensity: 63.08 mWmm⁻²), ChR2 ion channels within neurons in a specific target region of the cerebral cortex of the transgenic mouse are depolarized without overheating the adjacent local tissue, leading to the generation of action potentials and subsequent activation of neurons (Supplemental Figure S34).⁶⁴

Line 642:

Ray-tracing simulation

To perform a comprehensive 3D ray-tracing simulation based on the Monte-Carlo method, commercial software called OpticStudio 16.0 by ZEMAX, Inc. was utilized. The simulation employed a monochromatic point light source with a cone angle of 19.5°, inducing TIR in the waveguide (Supplemental Figure S8), located at the entrance of the waveguide structure. The refractive index of the waveguide was set to 1.47, and for increased accuracy, the refractive indices of the Si and metal were obtained from measured results.^{65,66} The simulation used 1×10^7 rays to obtain stable calculation results, and a rectangular detector with dimensions of 2500 × 500 pixels in the x and y directions was employed to record the light propagation from the entrance to the tip of the waveguide. 2D light distributions were reconstructed using MATLAB software by Mathworks Inc.

Path analysis in OpticStudio is conducted to specify absorption loss at the anti-artifact layer. The result of the path analysis is graphically represented using Origin 2022 by OriginLab Corporation.

There is an additional supplementary figure.

Supplementary Figure S7 | The single-channel waveguide simulation with Mo reflective layer. (a) Simulation system to measure loss of single-channel waveguide. The waveguide consists of PLGA waveguide, PLGA substrate, Mo reflective layer and Si nanomembrane. The inset below depicts rays which propagate within the waveguide at various angle. (b) The loss of each discretized angle and cumulative loss. A larger incident angle results in great increased losses due to absorption. The cumulative loss is calculated by adding up the power contributions of each angle. (c) The number of rays in relation to the number of reflections on Mo. The inset illustrates how the number of reflections varies for rays starting at different incident angles.

Supplementary Figure S8 | The actual 4-channel waveguide simulation with Mo reflective layer. (a) Simulation system and detector arrangement for loss measurement. The blue segment at the end of waveguide is an extension of short waveguide to compensate the length mismatch. (b) Loss of the compensated waveguide. Absorption losses occur after where the waveguide overlaps with the Mo layer, indicated green area.

Comment #4: Lifetime of current device structure is 2 months, but the author should provide

further clarification on the duration of reliable operation in the body, particularly since the material used is biodegradable.

Our response: We appreciate the reviewer for raising this important comment. We conducted two additional experiments to verify the operational lifespan and residual lifespan of the device in a state of chronic *in vivo* implantation. After fully implanting the device into the cerebral cortex of transgenic mice through craniotomy, regular assessments were conducted to ensure that the device was operating as intended. These evaluations included monitoring the device's functionality, stability, and performance over an extended period. First, we observed the duration for which the device consistently maintained its initial functionality and performance. For chronic implantation in transgenic mice, we customized the headstage to integrate the optical cannula with the device. Subsequently, we conducted optogenetic stimulation and LFP recording in mice with chronic implants. The device maintained its initial performance and operated in a stable manner, recording similar evoked LFPs under the same stimulation conditions for more than 14 days following the initial implantation. Starting with malfunctions of the device on day 20, several channels exhibit signs of malfunction by day 21, due to the occurrence of an open-circuit state of the electrodes caused by bioresorption. Furthermore, the remaining operational channels exhibit reduced light transmission efficiency due to the *in vivo* biodegradation and erosion of the PLGA waveguide, leading to a reduction in the peak amplitude of the recorded LFPs to below 150 μ V attributed to the diminished light intensity for stimulation. Consequently, all channels failed by 28 days post-implantation.

Next, to further evaluate the biodegradable kinetic of the device, we utilized CT scanning. For 7 weeks, we observed the gradual breakdown and absorption of the device material, confirming its biodegradable nature. Through these combined efforts, we were able to determine both the functional lifespan of the device and its biodegradation duration during chronic implantation. The implanted device was traceable on day 21, after which it completely dissolved by the day 50. In summary, these results indicate that the fully implanted bioresorbable hybrid device operates temporarily while maintaining performance, successfully records evoked LFPs, and ultimately dissolves and disappears within the body.

Our modification to the manuscript:

There is a modification in the contents of the sentence.

Line 506: **Fig. 5f** display the evoked LFPs recorded from each electrode channel of the bioresorbable hybrid system, induced by 460 nm pulses (intensity: 63.08 mWmm⁻², duration: 30 ms, frequency: 2 Hz stimulation). On day 0, in contrast to the control group of wild-type mice, the peak of the optogenetically-induced evoked LFPs in the cerebral cortex of Thy-1:ChR2 transgenic mice occurred approximately 15 ms after initiation of stimulation, with magnitudes ranging from 200 to 250 μ V (Supplemental Figure S35 and S36). Additionally, as the stimulation intensity increased from 15.77 mWmm⁻², 31.54 mWmm⁻², to 63.08 mWmm⁻², evoked LFP peaks of 100 μ V, 160 μ V, and 230 μ V, respectively, were observed (Supplemental Figure S37). The device operates in a stable manner without any significant change in performance, recording similar evoked LFPs in the same stimulation condition for 14 days following the initial implantation. Starting with malfunctions of the device on day 20, several channels exhibit signs of malfunction by day 21, due to the occurrence of an open-circuit state of the electrodes caused by bioresorption.³⁴ And the remaining operational channel exhibit a

decrease in light transmission efficiency due to the *in vivo* degradation and erosion of the PLGA waveguide, leading to a reduction in the peak amplitude of the recorded LFPs to below 150 μV attributed to the diminished light intensity for stimulation.⁶⁵ Consequently, all channels had failed by the 28 days post-implantation. Additionally, the dissolution process of the implanted device was monitored using computed x-ray tomography. The implanted device was traceable on day 21, after which it completely dissolved by the day 50 (Supplemental Figure S38). Collectively, these results suggest that the fully implanted bioresorbable hybrid device operated transiently while maintaining its performance, successfully recording evoked LFPs with minimal artifacts and simultaneously transmitting light from a laser source for optogenetic stimulation, ultimately dissolving and disappearing *in vivo*.

There is a modification in the figure 5.

There is an additional supplementary figure.

Supplementary Figure S38 | Dissolution characteristic of the implanted device. Computed tomography images of coronal section of the mouse skull collected over 49 days following the device implantation. The white dotted box highlights the bioresorbable device.

Comment #5: Figure 5f is not very intuitive. The labeling of evoked LFP in the spectrogram should be considered.

Our response: We thank the reviewer for this comment. To avoid confusion of the readers, we have indicated the marker at which the local field potential is evoked both manuscript and figure.

Our modification to the manuscript:

There is a modification in the contents of the sentence.

Line 501: **Fig. 5e** shows the spectrogram of evoked LFP power spectral density in the frequency range from 0 to 300 Hz. The dotted lines highlight the time point when the LFP activities are evoked. The evoked local field potentials were particularly strong at frequencies below 30Hz and exhibited periodicity, extending up to 300Hz.

There is a modification in the figure 5.

Reviewer #2

General Comment #1: This study presents a neural interface focused on the development of a bioresorbable, flexible hybrid opto-electronic system for simultaneous electrophysiological recording and optical stimulation. This proposed device addresses the limitations of current bioabsorbable neural implants, which typically perform only single functions. The system is composed of biodegradable PLGA and demonstrates biocompatibility, enabling direct optical and electrical interfacing with the curved cerebral cortex surface. The device has been optimized to minimize light transmission losses and photoelectric artifact interference, and its effectiveness has been demonstrated through in vivo experiments with transgenic mice. The authors highlight the potential of this novel hybrid neural implant system for monitoring neural activities and providing therapeutic interventions in various biomedical applications. The primary critiques of this study emphasize the need for a more comprehensive evaluation and characterization of the as-fabricated devices, both pre- and post-implantation. While the research presents valuable contributions to the field of neural interface development, the limitations and concerns raised suggest that further investigation is necessary to establish this work as a significant advancement in the field.

Our response: We thank the reviewer for positive comments. We sincerely accommodated the reviewer's opinion and revised our manuscript. Details of the revisions are provided in Our response.

Major comment #1: The authors claimed that the entire device is bioresorbable, and the lifetime of the device can be adjusted by controlling the materials parameters such as thickness, doping concentration of Si, and the composition ratio of PLGA. Also, an insulating layer composed of silicon dioxide (SiO₂) was used. However, questions remain regarding the integrity and performance of the encapsulation, and the stability of the entire device, particularly in the context of potential current leakage that may arise as the device degrades over time. To address these concerns, it is essential to provide a comprehensive analysis of the electrochemical properties and electrical insulation under various physiological conditions over the degradation.

Our response: We thank the reviewer for considerate comment. In order to provide readers with more comprehensive and clear information, we conducted additional in vitro experiments to measure the performance and lifespan of the silicon dioxide (150nm) barrier used in the device. The biodegradable device has both an operational lifespan during which it functions normally and a lifespan during which it completely decomposes and disappears. The silicon dioxide barrier is used as an insulating layer to ensure the stable operation of the device. The measurements were performed under conditions that simulate a physiological environment, specifically in PBS (Phosphate Buffered Saline) solution at 37°C. As a result, the encapsulation layer functioned as a stable insulating layer for 20 days before developing shorts with the external environment.

Our modification to the manuscript:

There is a modification and addition in the contents of the sentence.

Line 323: After implantation, the overall stability of the device is ensured for a period of 3 weeks and then operation may become restricted due to the degradation of the SiO₂ insulation layer (Supplemental Figure S10). Subsequently, the device will take 2 months for completely dissolved, and the lifetime of the device can be adjusted by controlling the material parameters such as thickness, doping concentration of Si, and composition ratio of PLGA by programming manners.⁴²

There is an additional supplementary figure.

Supplementary Figure S10 | Prolonged stability test of SiO₂ encapsulation layer. Electrochemical impedance changes of a Mo/Si electrode with a 150nm-thick SiO₂ encapsulation layer were observed in a phosphate-buffered saline solution at pH 7.4 at 37 °C over a period of 25 days.

Major comment #2: Connected with the previous comment, why was SiO₂ employed for the insulation and what are the merits of SiO₂ over other biodegradable insulating polymers? It would be valuable to explore alternative biocompatible and bioresorbable insulating materials that could potentially mitigate the risk of current leakage and provide enhanced encapsulation performance. Comprehensive comparative studies on the durability and biocompatibility of various insulating materials could potentially identify alternatives to SiO₂ for bioresorbable device applications.

Our response: We thank the reviewer for the insightful advice. Biodegradable devices inherently have limitations in material selection, which restricts the range of choices for materials and processing methods in device fabrication. Various polymers such as poly(octamethylene maleate (anhydride) citrate) (POMaC), poly(glycerol sebacate) (PGS), polyanhydride (PA), and poly(lactide-co-glycolide) (PLGA) are used as organic insulating films. However, due to their low glass transition temperature and high solubility in organic solvents, biodegradable polymers are not compatible with conventional fabrication processes such as photo-lithography, wet or dry etching. Additionally, inorganic encapsulants allow for the fabrication of thinner devices at the same performance, as they have relatively lower water vapor transmission rates (WVTR) compared to organic encapsulants. In our proposed biodegradable device integrating PLGA waveguide and ECoG electrodes, the electrodes for electrophysiological recordings are transferred onto the PLGA substrate topped with a waveguide in the final step of the device fabrication process. Therefore, SiO₂ encapsulation

provides a more secure level of device protection compared to organic materials, and it is more compatible with our device process.

Therefore, before the integration of the waveguide and electrode array, our ECoG electrode array is fabricated on a wafer substrate using standard semiconductor processes, allowing us to apply inorganic silicon dioxide as the insulating layer. To eliminate reader confusion, we evaluated the performance of the SiO₂ insulating film, a biodegradable inorganic insulator, through in vitro tests in an artificial physiological environment, and added references to other biodegradable inorganic and organic insulators.

Our modification to the manuscript:

There is a modification and addition in the contents of the sentence.

Line 287: This inorganic thin-film encapsulation layer on the device offers higher encapsulation performance with lower water vapor transmission rates compared to organic-based encapsulation layers, ensuring stable device protection even with its thin thickness.⁵⁰⁻⁵³

There is an addition in the references.

- 50 Boutry, C. M. *et al.* A stretchable and biodegradable strain and pressure sensor for orthopaedic application. *Nature Electronics* **1**, 314-321 (2018).
- 51 Boutry, C. M. *et al.* Biodegradable and flexible arterial-pulse sensor for the wireless monitoring of blood flow. *Nature biomedical engineering* **3**, 47-57 (2019).
- 52 Choi, Y. S. *et al.* Fully implantable and bioresorbable cardiac pacemakers without leads or batteries. *Nature biotechnology* **39**, 1228-1238 (2021).
- 53 Lee, G. *et al.* A bioresorbable peripheral nerve stimulator for electronic pain block. *Science Advances* **8**, eabp9169 (2022).

Major comment #3: The authors demonstrated the achievement of total internal reflection within the device by establishing a refractive index contrast and presenting ray tracing simulation results based on the refractive index values. However, given different fractions of polymers exhibit different refractive indices, the precise characterization of the refractive index for the specific PLGA formulation used in this study remains unclear and warrants further elaboration. The authors should consider conducting experimental measurements of the refractive index of the PLGA formulation with different concentrations, detailing the methodology and results in the manuscript.

Our response: We thank the reviewer for the sharp comment. During the process of selecting the composition ratio of glycolic acid and lactic acid in PLGA, we fabricated PLGA polymer films with different composition ratios of 75:25 and 50:50. We measured the refractive index and found that the difference was negligible, in below table. So, for more stable operating after in-vivo implantation, we opted for PLGA with a composition ratio of 75:25, which has a more stable and slower degradation rate. Additionally, detailed information about the PLGA used has been added to the Methods section.

Sample	404nm	532nm	638nm	829nm
PLGA 75:25 #1	1.4752	1.4660	1.4605	1.4539
PLGA 75:25 #2	1.4737	1.4629	1.4587	1.4546
PLGA 75:25 #3	1.4761	1.4670	1.4608	1.4532
PLGA 50:50 #1	1.4784	1.4696	1.4640	1.4572
PLGA 50:50 #2	1.4791	1.4676	1.4622	1.4563
PLGA 50:50 #3	1.4785	1.4705	1.4647	1.4573
PLGA 75:25 average	1.4750	1.4653	1.4600	1.4539
PLGA 50:50 average	1.4787	1.4692	1.4636	1.4569

Our modification to the manuscript:

There is a modification and addition in the contents of the sentence.

Line 292: The light entering the waveguide, is guided through the waveguide by inducing TIRs at the interface between the cerebrospinal fluid (CSF) ($n=1.37$) and PLGA ($n=1.47$) (Supplemental Figure S6).

Line 606:

Refractive index measurement

Prepared each PLGA-chloroform solution (5%w/v) (PLA:PGA=75:25, $M_w=85,000$ g/Mol, $IV=0.85$ and PLA:PGA=50:50, $M_w=90,000$ g/Mol, $IV=0.90$ dL/g, Rimless Industry Co.) was poured onto a Teflon dish (diameter: 2inch) and dried at room temperature to create 100 μm thick PLGA films. Films cut to 8x8mm were loaded into the prism coupler(2010/M, METRICON), and the refractive index was measured.

There is an additional supplementary figure.

Wavelength	404 nm	532 nm	632.8 nm	829 nm
$n_{\text{PLGA 75:25}}$	1.4750	1.4653	1.4600	1.4539
$n_{\text{PLGA 50:50}}$	1.4787	1.4692	1.4636	1.4569

(Thickness of film : 100 μm , 23°C, $n=3$)

Supplementary Figure S6 | PLGA film refractive index as a function of composition.

Major comment #4: Following the previous comment, the authors presented the transmittance measurements of Mo/Si, Mo, and Si nanomembranes in the visible light range. However, the absorbance/transmittance and autofluorescence properties of the PLGA waveguide were not discussed. Furthermore, the molecular weight of PLGA and its lactic-to-glycolic acid ratio was not mentioned in the methods section, which is a crucial detail, as variations in these parameters can influence the autofluorescence of the polymer. Additionally, the fabrication process involves soft baking above 95°C, which may contribute to increased autofluorescence in the material. It is essential to provide a comprehensive analysis of the absorbance/transmittance and autofluorescence properties of the PLGA waveguide to fully evaluate the device's

performance.

Our response: We appreciate the reviewer for the important comment. We have added accurate and detailed information about the PLGA used in the device, along with measured data on optical transmittance in **Fig S2**.

Supplementary Figure S2 | Visible light transmittance spectra of the PLGA film.

The manufacturing processes for the PLGA waveguide and Si-based electrodes are independent. The PLGA waveguide is made by filling a pre-fabricated PDMS mold with a 5% w/v PLGA solution, and then evaporating it at room temperature. On the other hand, the ECoG electrodes are manufactured on a wafer using standard semiconductor processes. Ultimately, the independently produced waveguide and electrodes are integrated at room temperature through a transfer printing method. Because PLGA remains at room temperature throughout the entire manufacturing process, and no baking step is involved, it is possible to eliminate any autofluorescence enhancement that might occur due to baking. To avoid confusing the reader, we have organized each step of the process under separate subheadings in the methods section.

Our modification to the manuscript:

There is a modification and addition in the contents of the sentence.

Line 280: The electrode array is constructed on a PLGA-based transparent substrate (20 μm) in which embossed waveguides (100 μm) are shaped through soft lithography (Supplemental Figure S2).

Line 559:

PLGA waveguide fabrication

Fabrication of the poly(lactic-co-glycolic acid)(PLGA) waveguide start with fabrication of master mold. On the prepared glass substrate, polyimide (PI) ($\sim 4 \mu\text{m}$) is spun coated, baked (150 $^{\circ}\text{C}$ for 10 min, 210 $^{\circ}\text{C}$ for 180 min) and Cu ($\sim 300 \text{ nm}$) is deposited by sputtering. After patterning the waveguide pattern through photolithography, PI and Cu are etched, sequentially. And spin-casting SU-8 100 ($\sim 100 \mu\text{m}$) on a glass substrate. Photoexposure of the SU-8 100 layer after soft baking (65 $^{\circ}\text{C}$ for 20 min, 95 $^{\circ}\text{C}$ for 90 min) was carried out for patterning waveguides in a state where the glass substrate was turned over and fixed zig to tilt the master mold at 60 $^{\circ}$ (Supplemental Figure SX). After post exposure baking (65 $^{\circ}\text{C}$ for 1 min, 95 $^{\circ}\text{C}$ for 15 min) and SU-8 100 development, the fabricated master mold was fixed on a petri dish and 5:1 polydimethylsiloxane (PDMS, Dow Corning) was poured to obtain a PDMS mold. The

completely cured PDMS mold was separated from the master mold and petri dish before being fixed on another glass substrate with quicksilver to prevent deformation of the mold. A fabricated PLGA-chloroform solution (5%w/v) (PLA:PGA=75:25, Rimless Industry Co., Mw=85,000 g/Mol, IV=0.85 dL/g) was poured over the PDMS mold and dried at room temperature to form the PLGA waveguide film. The fabricated dry film was not separated from the PDMS mold before electrode array transfer printing.

Mo/Si electrode array fabrication

Fabrication of the electrode array started with solid-state phosphorus doping (PH-1000N Source, Saint Gobain, 950 °C for 15 min) of a silicon-on-insulator (SOI, top Si ~300 nm, SOITEC) wafer. We separated the Si nanomembrane (SiNM) on the PDMS slab by wet etching the buried oxide layer of the SOI wafer with concentrated HF and transfer-printed on a Si wafer substrate upon which poly(methylmethacrylate) (PMMA, ~1 µm) and polyimide (PI, ~150 nm) were spin cast sequentially. A Mo layer (~300 nm) was deposited through e-beam evaporation and the device pattern was defined through photolithography and reactive ion etching (RIE) for Mo, Si NM patterning. The PI top layer (~150 nm) was spin cast and patterned for accessibility of acetone for PMMA dissolution. The multilayer PI, Si, Mo, and PI was separated from the Si substrate by PMMA dissolution in acetone and retrieved reversely from the Si wafer for bottom PI layer etching by RIE. The PI, Mo, and Si multilayer on the Si wafer was stamped using a PDMS stamp, and the top PI layer was etched by RIE. The remaining Si, Mo multilayer was transferred to another PMMA (~1 µm), PI (~150 nm) spin-cast Si wafer substrate. The order of the above processes was to change the order of the Si and Mo layers. A layer of SiO₂ (~150 nm) was formed through plasma-enhanced chemical vapor deposition. Buffered oxide etchant was used to remove the SiO₂ layer from the electrode regions, and a top PI layer (~150 nm) was spin cast and patterned for a neutral mechanical plane.

Device integration

The fabricated electrode array was separated from the Si wafer substrate by PMMA dissolution in acetone, and the retrieved Si wafer was used for top PI layer etching by RIE. The electrode array was stamped with a PDMS stamp and bottom PI layer was etched by RIE. The PDMS slab with the electrode array was stamped with the PLGA waveguide film whose surface is lightly swollen by an acetonitrile solvent to remove the PDMS slab after drying. We completed the integration by separating the PLGA waveguide film with electrode array and PMDS master mold. Then, ACF cables were bonded to the via regions of the electrode array interconnects for connection to an external data acquisition (DAQ) system. Additionally, fiber optic cannula (Ø1.25 x 6.4 mm ceramic ferrule, Ø105 µm core, 0.22 NA, l=2 mm, Thorlabs) were precisely aligned to the entrance of the PLGA waveguides on the customized head-stage made by 3D printing and bonded using Norland Optical Adhesive 76 (NOA 76, Norland Products).

There is an additional supplementary figure.

Supplementary Figure S2 | Visible light transmittance spectra of the PLGA film.

Major comment #5: Bending tests were performed on devices wrapped around a cylindrical glass rod to demonstrate the flexibility and biocompatibility of devices. But it is not enough to evaluate the performance of devices after implantation since the mechanical properties (i.e., young's modulus and elongation) are unknown in this study. Mechanical testing of the devices (i.e., tensile tests) should be included in this study. Additionally, the degradation of device in vivo was not thoroughly examined. Does the device cause any immune response or scar formation during implantation and degradation at the implantation site? A comprehensive assessment should be carried out to determine whether the device elicits any immune responses or scar formation during implantation and degradation at the implantation site. This information is important to neuroscientists and other researchers who may be interested in utilizing these devices.

Our response: We thank the reviewer for the thoughtful comments. Our device is a combination of polymer waveguides and inorganic nano-membrane electrode array. As shown in **Fig. S1**, we measured and added data on the modulus of elasticity for the PLGA polymer substrate, which predominantly determines the mechanical properties of the device. However, our device is not stretchable but is flexible, owing to the inclusion of inorganic thin films, specifically silicon films. To evaluate the device's performance through tensile testing, it's necessary to assess the electrodes' functionality at strains lower than the fracture strain of silicon nano-membrane which is generally less than 1%. Therefore, instead of conducting durability tests through elongation, we carried out mechanical tests under bending and repeated bending conditions.

Supplementary Figure S1 | Stress-strain curves of the PLGA film. (a) Elastic-plastic property of PLGA film. (b) Calculated Young's modulus of PLGA film by linear fitting.

For immune responses, we examined activated astrocyte through glial fibrillary acidic protein (GFAP) and activated macrophages and microglia through ionized calcium-binding adaptor molecule 1 (Iba-1) in the coronal section of somatosensory cortex. Immunohistochemistry was performed at 8-week post-implant. although overall GFAP activity increased at the implantation site (ipsilateral hemisphere), we found little difference in glial activities of implantation site compared to contralateral hemisphere. Thus, we suggest that the high biocompatibility of the materials employed in the device.

Next, we conducted two additional experiments to verify the operational lifespan and residual lifespan of the device in a state of chronic *in vivo* implantation. After fully implanting the device into the cerebral cortex of transgenic mice through craniotomy, regular assessments were conducted to ensure that the device was operating as intended. These evaluations included monitoring the device's functionality, stability, and performance over an extended period. First, we observed the duration for which the device consistently maintained its initial functionality and performance. For chronic implantation in transgenic mice, we customized the headstage to integrate the optical cannula with the device. Subsequently, we conducted optogenetic stimulation and LFP recording in mice with chronic implants. The device maintained its initial performance and operated in a stable manner, recording similar evoked LFPs under the same stimulation conditions for more than 14 days following the initial implantation. Starting with malfunctions of the device on day 20, several channels exhibit signs of malfunction by day 21, due to the occurrence of an open-circuit state of the electrodes caused by bioresorption. Furthermore, the remaining operational channels exhibit reduced light transmission efficiency due to the *in vivo* biodegradation and erosion of the PLGA waveguide, leading to a reduction in the peak amplitude of the recorded LFPs to below 150 μ V attributed to the diminished light intensity for stimulation. Consequently, all channels failed by 28 days post-implantation.

Next, to further evaluate the biodegradable kinetic of the device, we utilized CT scanning. For 7 weeks, we observed the gradual breakdown and absorption of the device material, confirming its biodegradable nature. Through these combined efforts, we were able to determine both the functional lifespan of the device and its biodegradation duration during chronic implantation. The implanted device was traceable on day 21, after which it completely dissolved by the day 50. In summary, these results indicate that the fully implanted bioresorbable hybrid device operates temporarily while maintaining performance, successfully records evoked LFPs, and ultimately dissolves and disappears within the body.

Our modification to the manuscript:

There is a modification in the contents of the sentence.

Line 270: The device constructed with a nanomembrane electrode array transferred to a PLGA waveguide substrate is soft and flexible, minimizing the modulus mismatch between tissue and implant and allowing conformal contact with the tissue (Supplemental Figure S1).

Line 323: After implantation, the overall stability of the device is ensured for a period of 3 weeks and then operation may become restricted due to the degradation of the SiO₂ insulation layer (Supplemental Figure S10). Subsequently, the device will take 2 months for completely dissolved, and the lifetime of the device can be adjusted by controlling the material parameters such as thickness, doping concentration of Si, and composition ratio of PLGA by programming manners.⁴²

Line 460: Furthermore, immune responses were examined through immunohistochemistry. **Fig. 4c** and **d** display the results following the procedures outlined in the method section. Double immunostaining for astrocytes (GFAP) and microglia (Iba-1), in **Fig. 4c** and **d**, reveals glial cell activation 8 weeks post-implantation. In bioresorbable devices, moderate subdural gliosis occurs at the implantation site, when compared to the control contralateral hemisphere. As shown in **Fig. 4e**, notably, there was no significant difference in immune responses between the region of device placement and the control contralateral hemisphere. These results indicate the high biocompatibility of the materials used in the device, ensuring its stable performances.

Line 506: Line 506: **Fig. 5f** display the evoked LFPs recorded from each electrode channel of the bioresorbable hybrid system, induced by 460 nm pulses (intensity: 63.08 mWmm⁻², duration: 30 ms, frequency: 2 Hz stimulation). On day 0, in contrast to the control group of wild-type mice, the peak of the optogenetically-induced evoked LFPs in the cerebral cortex of Thy-1: ChR2 transgenic mice occurred approximately 15 ms after initiation of stimulation, with magnitudes ranging from 200 to 250 μV (Supplemental Figure S35 and S36). Additionally, as the stimulation intensity increased from 15.77 mWmm⁻², 31.54 mWmm⁻², to 63.08 mWmm⁻², evoked LFP peaks of 100 μV, 160 μV, and 230 μV, respectively, were observed (Supplemental Figure S37). The device operates in a stable manner without any significant change in performance, recording similar evoked LFPs in the same stimulation condition for 14 days following the initial implantation. Starting with malfunctions of the device on day 20, several channels exhibit signs of malfunction by day 21, due to the occurrence of an open-circuit state of the electrodes caused by bioresorbtion.³⁴ And the remaining operational channel exhibit a decrease in light transmission efficiency due to the *in vivo* degradation and erosion of the PLGA waveguide, leading to a reduction in the peak amplitude of the recorded LFPs to below 150 μV attributed to the diminished light intensity for stimulation.⁶⁵ Consequently, all channels had failed by the 28 days post-implantation. Additionally, the dissolution process of the implanted device was monitored using computed x-ray tomography. The implanted device was traceable on day 21, after which it completely dissolved by the day 50 (Supplemental Figure S38). Collectively, these results suggest that the fully implanted bioresorbable hybrid device operated transiently while maintaining its performance, successfully recording evoked LFPs with minimal artifacts and simultaneously transmitting light from a laser source for optogenetic stimulation, ultimately dissolving and disappearing *in vivo*.

There is a modification in the figure 4.

There is a modification in the figure 5.

There is an additional supplementary figure.

Supplementary Figure S1 | Stress-strain curves of the PLGA film. (a) Elastic-plastic property of PLGA film. (b) Calculated Young's modulus of PLGA film by linear fitting.

Supplementary Figure S38 | Dissolution characteristic of the implanted device. Computed tomography images of coronal section of the mouse skull collected over 49 days following the device implantation. The white dotted box highlights the bioresorbable device.

Major comment #6: The authors reported optogenetics powers in the range of 73.15 – 1463.04 mW/mm², it is helpful for evaluating the light-induced artifacts with such large powers, but the optogenetic stimulation threshold is way lower than 73.15 mW/mm². Such elevated power levels may inadvertently activate heat-sensitive channels in addition to ChR2, potentially leading to overheating of brain tissue and confounding the experimental results. The authors should measure and report the temperature at both the output of the devices and the implantation sites during optogenetic stimulation to ensure that tissue overheating is not a confounding factor in the study.

Our response: We thank the reviewer for the helpful comment. First, we have performed measurements on the intensity of the light emitted from the tip of the waveguide. The light transmission efficiency was quantified to verify the effectiveness of the PLGA waveguide in delivering light to the target tissue. Also, to enhance the reliability and precision of our device, we have additionally manufactured a headstage that was not previously implemented. The created headstage seamlessly connects the optical cannula with the device. When the output power of the optical cannula is 26.4 mW, the average light intensity emitted from the end of the waveguide to stimulate ChR2 is 63.06 mWmm⁻².

Next, to verify the thermal degradation of brain tissue due to laser irradiation, photostimulation was applied at 63.06mWmm⁻², 20Hz for 20ms, and the temperature change at the stimulated tissue was monitored with an IR camera. Temperature was monitored for 5

minutes under anesthesia, followed by another 5 minutes of monitoring while the tissue was stimulated. The results showed that there was a temperature increase of 0.1 degree just before and after stimulation. However, since there was also a 0.1 degree rise in temperature during the anesthetized state without stimulation, this appears to be a natural rise in body temperature, indicating that there is no thermal degradation of tissue due to the laser.

Our modification to the manuscript:

There is a modification and addition in the contents of the sentence.

Line 496: Upon exposure to 460 nm pulses transmitted from a waveguide (input power: 24 mW, stimulation intensity: 63.08 mWmm^{-2}), ChR2 ion channels within neurons in a specific target region of the cerebral cortex of the transgenic mouse are depolarized without overheating the adjacent local tissue, leading to the generation of action potentials and subsequent activation of neurons (Supplemental Figure S34).⁶⁴

There are additional supplementary figures.

Supplementary Figure S34 | Monitoring temperature of the cerebral cortex and the implanted device. (a) Plot of the temperature in the stimulation area before and during a 5-minute optical stimulation. The stimulation condition is a 460nm blue light pulse with an intensity of 63.08 mWmm^{-2} , a duration of 20 ms, and a frequency of 20 Hz. (b) Infrared thermal images collected at 2-minute intervals. The white crosshair aims the stimulation area.

Major comment #7: How to connect the device to the external light source?

Our response: We appreciate to important comment. In order to validate the usefulness of our device in living animals, we conducted revised *in vivo* experiments for chronic implantation.

To facilitate full implantation for these chronic tests, we manufactured an additional head stage, fabricated through 3D printing, ensures a snug fit for the optical cannula and device, and is affixed using the optical adhesive NOA76. Furthermore, an external laser is channeled through an optical fiber and connected to the cannula via an interconnector. The device with the I/O connector connected to the head stage was finally fully implanted by applying dental cement.

Our modification to the manuscript:

There is a modification and addition in the contents of the sentence.

Line 490: Similar to the *in vivo* ECoG recording, a biodegradable hybrid system equipped with a head-stage, ACF cable, and optical cannula was placed on an exposed $5 \times 8 \text{ mm}^2$ area of the cerebral cortex through craniotomy (Supplemental Figure S32 and S33). Details are explained in the Methods.

There are additional supplementary figures.

Supplementary Figure S32 | Head-stage design for device implantation. (a) 3D rendering model of head-stage. (b) A photograph of the head-stage and device integration with supporting zig.

Supplementary Figure S33 | Experimental setup for *in vivo* ECoG recording and optogenetic stimulation. (a) Photograph of the cranial window for *in vivo* recording and stimulation experiments before applying the dental cement. (b) Photograph after fully implanting the device.

Major comment #8: In figure 5b-c, if the authors claim that it is seizure-like spiking activity, please provide single-spike sorting.

Our response: We thank the reviewer for this comment. We randomly sampled 20 seizure-like spiking activities and sorted the spikes. The red line is the average value.

Our modification to the manuscript:

There is a modification and addition in the contents of the sentence.

Line 480: Seizure-like spiking activities induced by pilocarpine were recorded by the representative electrodes from both the bioresorbable and control electrode arrays, as presented in **Fig. 5c** and Supplemental Figure S29.

There is an additional supplementary figure.

Supplementary Figure S29 | single-spike sorting of recorded seizure-like spiking activity. Spikes (a) recorded from Mo/Si electrode and (b) recorded from Au electrode.

Minor comment #1: Please double confirm the optogenetic stimulation power listed in the main text (line 358, 367, 371, and 375), the units are inconsistent with the Supplementary Figure S12.

Our response: We thank the reviewer for the delicate comments. We have made all the changes as reviewer suggested.

Our modification to the manuscript:

There is a modification and addition in the contents of the sentence.

Line 410: To evaluate light-induced artifacts, 460 nm laser pulses (core diameter: 105µm, duration: 100ms, frequency: 2Hz) were applied to each electrode while gradually increasing the light intensity from 126.1 mWmm⁻² to 2522.5 mWmm⁻² through an optical fiber fixed perpendicular to the stereotaxis (Supplemental Figure S19).

Line 420: The artifacts with peaks from 400µV to 3mV were induced as the light intensity increased from 126.1 mWmm⁻² to 2522.5 mWmm⁻² and at higher light intensities, the artifacts

reached a saturation state (Supplemental Figure S23).

Line 423: In contrast, when the Mo/Si bilayer electrode was illuminated at intensities below 630.6 mWmm^{-2} , low levels of noise without any artifacts were observed. Negligible artifacts were observed below peak level of $200 \mu\text{V}$ with electrodes exposed to light intensities below 2522.5 mWmm^{-2} .

Line 427: Blue light of intensity 126.12 mWmm^{-2} was irradiated on the stimulation electrode, and blue light of intensity 73.15 mWmm^{-2} passed through the electrode to stimulate the tissue.

Minor comment #2: Based on the requirements of data presentation by the nature portfolio, please show individual points for the histograms when $n \leq 10$ (Figure 4d).

Our response: We appreciate the reviewer remind of the nature portfolio. We plotted individual points for the histogram. The histograms were modified with individual points in Supplemental Figure S27.

Our modification to the manuscript:

There is an additional supplementary figure.

Supplementary Figure S27 | Results of cell viability test. (a) Representative images indicating healthy condition of the cultured primary hippocampal neuronal cells on both the control and biodegradable electrode arrays. (b) Quantitative analyses of the live/dead assays on the control (blue) and bioresorbable (yellow) samples using Invitrogen™ EVOS™ digital color fluorescence microscope (Au: mean, s.e.m., Mo/Si: mean, s.e.m.) ($n = 12, 10$, where n is each sample number).

Minor comment #3: Please normalize the optogenetics power by the area, this may confuse the audience (line 429 and 439).

Our response: We thank the reviewer for helpful comment. To avoid confusion among readers, we have standardized all notation for stimulation conditions to power intensity (unit: mWmm^{-2}).

Our modification to the manuscript:

There is a modification and addition in the contents of the sentence.

Line 496: Upon exposure to 460 nm pulses transmitted from a waveguide (input power: 24 mW, stimulation intensity: 63.08 mWmm^{-2}), ChR2 ion channels within neurons in a specific target region of the cerebral cortex of the transgenic mouse are depolarized without overheating the adjacent local tissue, leading to the generation of action potentials and subsequent activation of neurons (Supplemental Figure S34).⁶⁴

Line 511: Additionally, as the stimulation intensity increased from 15.77 mWmm^{-2} , 31.54 mWmm^{-2} , to 63.08 mWmm^{-2} , evoked LFP peaks of 100 μV , 160 μV , and 230 μV , respectively, were observed (Supplemental Figure S37).

Minor comment #4: Please report the number of mice used for all *in vivo* experiments: (i) total number of mice utilized in the study. (ii) number of mice in each experimental group, along with any control groups. (iii) any instances of mice being excluded from the analysis, and the reasons for their exclusion. (iv) information on the randomization and blinding procedures employed, if applicable.

Our response: We used 23 mice for total *in vivo* experiments. For optic stimulation and ECoG recording, we used 8 transgenic and 2 wild-type mice (control). For photoinduced artifact evaluations, we used 2 transgenic and 2 wild-type mice. For CT imaging, we used 5 wild-type mice. For immunostaining, we used 1 transgenic and 2 wild-type mice. For temperature monitoring, we used 1 wild-type mouse. If *in vivo* signal is well-recorded, we terminated *in vivo* experiment and analyze the data. We were strictly adhered to during all mouse-related experiments.

Our modification to the manuscript: None.

Minor comment #5: In the Conclusion: (i) please include the limitations or future improvements of the as-fabricated bioresorbable devices since they require the sophisticated photolithography processing which is costly and time consuming. (ii) the throughput of the devices is unknown, please discuss the possibility in scaling up these bioresorbable devices since ECoG arrays have been augmented to about 1000 channels by other researchers.

Our response: We thank the reviewer for insightful advice. In the conclusion section, we discussed several specific strategies regarding the potential challenges and improvements of the device. In addition to addressing cost-effectiveness and the scalability of the device, as reviewer mentioned, we also examined considerations necessary for future clinical applications such as, control the biodegradation kinetics and integration with wireless systems.

Our modification to the manuscript:

Line 550: Additionally, its biodegradable nature mitigates the risk of long-term complications inherent to traditional implants and simplifies the device extraction process, which is anticipated to bring substantial improvements to clinical methodologies. Moving forward, synergistic advancements across various engineering domains will enable enhanced cost-effectiveness in the manufacturing process, precise control of biodegradation kinetics, scaling

up to high-density and large-area formats, and integration with wireless systems, collectively making it a more favorable choice for future clinical applications.

Reviewer #3

General Comment #1: The paper proposes a resorbable flexible opto-electronic system for simultaneous electrophysiological recording and optical stimulation. In vivo recording is achieved with photo-stimulation on transgenic mice. The paper should have broad interests in the community of neuroscience and neuroengineering. To improve the paper, I have the following comments.

Our response: We thank the reviewer for all the comments. Also, we appreciate the reviewer for summing up our manuscript and presenting some issues to be addressed. We did our best to reflect all the reviewer's comments. The corresponding revisions to each comment are indicated in response.

Major comment #1: Fig. 3a, the authors propose a Mo coated Si electrode design to eliminate the artefacts. At first glance, this seems feasible. However, the authors should note that this design can only block the illumination on the top surface of Si, while a considerable amount of photons can be backscattered by the tissue and enter the bottom side of Si, inevitably creating artefacts. Besides measurements, optical simulations should be performed to compare the number of photons captured by the Si film with and without Mo coatings.

Our response: We thank the reviewer for the astute observation regarding the potential for backscattered light to re-enter the Si film from the tissue, which may indeed result in artifacts. To explore the impact of backscattered light within the tissue, we utilized the Henyey-Greenstein bulk scattering model to simulate scattering events within the tissue[R1, R2]. As depicted in **Fig. S26a**, we introduced ray detectors on both the upper and lower surfaces of the Si film to quantify photon absorption from the waveguide, as well as backscattered light emanating from the tissue, respectively.

a.

b.

c.

Supplementary Figure S26 | Results depicting photon absorption with and without Mo reflective layer. (a) Schematic representation of the simulation setup with two ray detectors positioned at the upper and lower surfaces of the Si film to measure the amount of photon absorption from both directions. (b) Contour plots for absorption profiles with/without Mo anti-artifact layer captured by ray detectors, indicating photon incidence from waveguide and backscattering from tissue. Si nanomembranes are highlighted by white dashed lines and PLGA waveguide is denoted by red dashed lines. (c) Proportion of photons absorbed or transmitted within the system with/without Mo anti-artifact layer. The blue region indicating the total absorption of Si nanomembrane.

Fig. S26b presents the overall light profiles on the upper and lower surfaces of the Si film. By comparing the incident profiles from the top, a notable difference in photon incidence within the Si film is discerned (indicated by white dashed lines). With the Mo reflective layer in place, there is no absorption occurring on the Si film; however, without the Mo layer, a significant number of photons are absorbed where the waveguide (indicated by dashed red lines) overlaps with the Si film. Hence, the Mo reflective layer significantly mitigates direct photon absorption deemed as artifacts. The backscattered photons are predominantly observed near

the tip of the waveguide, with the absolute quantity being comparable in both cases. For a more granular analysis of backscattered photons, we delineated the proportion of total photons in **Fig. S27c**. With the Mo reflective layer, the reflection and absorption properties of the Mo layer result in the Si film absorbing only a minimal number of photons. Here, absorbed photons account for 0.659% of the total power, with 0.603% attributed to backscattering. Given this minuscule proportion, the influence of backscattering from the brain is deemed negligible. Conversely, without the Mo reflective layer, the Si film directly absorbs photons from the waveguide, amounting to 40.134% (*i.e.*, sum of absorption by Si top and back sides) of the total input power. In this scenario, the backscattered light, constituting merely 0.604% of the total, does not significantly contribute as the artifacts generated from the top layer absorption become dominant. The high absorption efficiency of Si for the operational wavelength substantially diminishes the absorption of backscattered light from the tissue.

[R1] Binzoni, T., Leung, T. S., Gandjbakhche, A. H., Rufenacht, D., & Delpy, D. T. (2006). The use of the Henyey–Greenstein phase function in Monte Carlo simulations in biomedical optics. *Physics in Medicine & Biology*, 51(17), N313.

[R2] Prahl, S. A. (1989, January). A Monte Carlo model of light propagation in tissue. In *Dosimetry of laser radiation in medicine and biology* (Vol. 10305, pp. 105-114). SPIE.

Our modification to the manuscript:

There is a modification and addition in the contents of the sentence.

Line 443: The ray-tracing simulation yielded consistent result with the previous experiment (Supplemental Figure S26). The presence of the Mo anti-artifact layer led to a remarkable reduction in photon absorption on the Si layer. Compared with the dominant absorption from result of the Si film without the anti-artifact layer, artifacts from backscattering by the tissue are not substantial enough to have a significant impact to the recorded signal.

Line 642:

Ray-tracing simulation

To perform a comprehensive 3D ray-tracing simulation based on the Monte-Carlo method, commercial software called OpticStudio 16.0 by ZEMAX, Inc. was utilized. The simulation employed a monochromatic point light source with a cone angle of 19.5° , inducing TIR in the waveguide (Supplemental Figure S8), located at the entrance of the waveguide structure. The refractive index of the waveguide was set to 1.47, and for increased accuracy, the refractive indices of the Si and metal were obtained from measured results.^{65,66} The simulation used 1×10^7 rays to obtain stable calculation results, and a rectangular detector with dimensions of 2500×500 pixels in the x and y directions was employed to record the light propagation from the entrance to the tip of the waveguide. 2D light distributions were reconstructed using MATLAB software by Mathworks Inc.

Path analysis in OpticStudio is conducted to specify absorption loss at the anti-artifact layer. The result of the path analysis is graphically represented using Origin 2022 by OriginLab Corporation.

a.

b.

c.

Supplementary Figure S26 | Results depicting photon absorption with and without Mo reflective layer. (a) Schematic representation of the simulation setup with two ray detectors positioned at the upper and lower surfaces of the Si film to measure the amount of photon absorption from both directions. (b) Contour plots for absorption profiles with/without Mo anti-artifact layer captured by ray detectors, indicating photon incidence from waveguide and backscattering from tissue. Si nanomembranes are highlighted by white dashed lines and PLGA waveguide is denoted by red dashed lines. (c) Proportion of photons absorbed or transmitted within the system with/without Mo anti-artifact layer. The blue region indicating the total absorption of Si nanomembrane.

Major comment #2: Fig. 3e (left, 20 mW) and Fig. 5e, the shapes of these recorded signals are very much alike, which makes one doubt that the results in Fig. 5e are artefacts, just as those in Fig. 3e (left, 20 mW). These possible artefacts can be ascribed to the reason mentioned in Comment 1. At least two additional in vivo experiments should be performed: (1) Do the experiments in Fig. 5d with wild-type mice; (2) In Fig. 5d, only illuminate one or two

waveguide channels, and measure the LFP signals from all the four electrode channels.

Our response: We thank the reviewer for insightful advice. As suggested, to convey our findings more clearly to the readers, we have revised the figures and conducted additional *in vivo* experiments. Previously, Figure 3f displayed results from the most intense stimulation (2522 mWmm^{-2}). However, since transgenic mice are actually stimulated with an intensity of 63 mWmm^{-2} during *in vivo* experiment, we have replaced the data of the artifact generation from Si electrodes and Mo/Si electrodes tested in wild-type mice with results from artifacts generated under stimulation at a fiber optic output of 126 mWmm^{-2} (stimulation intensity: 73 mW/mm^2), which resembles the *in vivo* experimental conditions. Previously, Mo/Si electrodes were shown to produce photoinduced artifacts, but under stimulation conditions similar to those in transgenic mice *in-vivo* test, any photoinduced artifacts were not observed. Additionally, under conditions of harsh and strong stimulation, artifacts may be produced due to backscattering, which is presented in Supplemental Figure S26. Nevertheless, as shown in Supplemental Figure S27, in simulation results, only a small number of photons are scattered, impacting the silicon surface, thereby indicating that the Mo/Si electrodes are not significantly impacted by backscattering-induced artifacts under typical light intensities.

Subsequently, as illustrated in Figure 5f, we fully implanted the device in the brain of transgenic mouse and conducted stimulation and recording simultaneously. The same procedure was also carried out in wild-type mice under identical conditions. As can be seen in Supplemental Figure S34, only spontaneous brain waves were recorded in the wild-type mouse, while induced LFPs were recorded in the transgenic mouse. So, we verified that the recorded evoked potentials are not photoelectric artifacts.

Our modification to the manuscript:

There is a modification and addition in the contents of the sentence.

Line 430: Blue light of intensity 126.12 mWmm^{-2} was irradiated on the stimulation electrode, and blue light of intensity 73.15 mWmm^{-2} passed through the electrode to stimulate the tissue. The control electrode was placed 1 mm away from the stimulation electrode. As shown in **Fig. 3f**, when light was illuminated on the Si stimulation electrode, artifacts with a peak level of $400 \mu\text{V}$ were observed, and artifacts with a peak level of $50 \mu\text{V}$ were also observed on the control electrode owing to light scattering by the tissue. On the other hand, the Mo/Si stimulation and control electrodes did not exhibit any artifacts. When stimulating brain tissue using blue light pulses with an intensity of $2522.48 \text{ mWmm}^{-2}$, artifacts with a peak level of $200 \mu\text{V}$ were observed on the Mo/Si stimulation electrode, which was induced by the scattered and reflected light at the surface of the brain tissue, and only low-noise brain activities were recorded without artifacts at the control electrode (Supplemental Figure S25).

Line 443: The ray-tracing simulation yielded consistent result with the previous experiment (Supplemental Figure S26). The presence of the Mo anti-artifact layer led to a remarkable reduction in photon absorption on the Si layer. Compared with the dominant absorption from result of the Si film without the anti-artifact layer, artifacts from backscattering by the tissue are not substantial enough to have a significant impact to the recorded signal.

Line 506: **Fig. 5f** display the evoked LFPs recorded from each electrode channel of the bioresorbable hybrid system, induced by 460 nm pulses (intensity: 63.08 mWmm^{-2} , duration: 30 ms, frequency: 2 Hz stimulation). On day 0, in contrast to the control group of wild-type mice, the peak of the optogenetically-induced evoked LFPs in the cerebral cortex of Thy-1:ChR2 transgenic mice occurred approximately 15 ms after initiation of stimulation, with magnitudes ranging from 200 to 250 μV (Supplemental Figure S35 and S36). Additionally, as the stimulation intensity increased from 15.77 mWmm^{-2} , 31.54 mWmm^{-2} , to 63.08 mWmm^{-2} , evoked LFP peaks of 100 μV , 160 μV , and 230 μV , respectively, were observed (Supplemental Figure S37). The device operates in a stable manner without any significant change in performance, recording similar evoked LFPs in the same stimulation condition for 14 days following the initial implantation. Starting with malfunctions of the device on day 20, several channels exhibit signs of malfunction by day 21, due to the occurrence of an open-circuit state of the electrodes caused by bioresorbtion.³⁴ And the remaining operational channel exhibit a decrease in light transmission efficiency due to the *in vivo* degradation and erosion of the PLGA waveguide, leading to a reduction in the peak amplitude of the recorded LFPs to below 150 μV attributed to the diminished light intensity for stimulation.⁶⁵ Consequently, all channels had failed by the 28 days post-implantation. Additionally, the dissolution process of the implanted device was monitored using computed x-ray tomography. The implanted device was traceable on day 21, after which it completely dissolved by the day 50 (Supplemental Figure S38). Collectively, these results suggest that the fully implanted bioresorbable hybrid device operated transiently while maintaining its performance, successfully recording evoked LFPs with minimal artifacts and simultaneously transmitting light from a laser source for optogenetic stimulation, ultimately dissolving and disappearing *in vivo*.

There is a modification in the figure 3.

There is a modification in the figure 5.

There are additional supplementary figures.

Supplementary Figure S25 | In vivo recorded artifact signals from Mo/Si electrodes with blue light illumination of intensity at $2522.48 \text{ mWmm}^{-2}$.

Supplementary Figure S35 | Recorded ECoG from the cerebral cortex of the transgenic mouse upon stimulation with 460 nm blue light pulses(intensity: 63.08mWmm^{-2} , frequency: 2 Hz, duration: 30ms).

Supplementary Figure S36 | Recorded ECoG from the cerebral cortex of the wild-type mouse upon stimulation with 460 nm blue light pulses(intensity: 63.08mWmm^{-2} , frequency: 2 Hz, duration: 30ms).

Major comment #3: Mo has very poor reflectivity, which is only ~50% at wavelengths of 400-

500 nm. There will be 50% loss at a single bounce! Guess how much power will be lost after multiple bounces? The waveguide loss (unit: dB/cm) should be measured and simulated.

Our response to comment #3: We appreciate the reviewer for the insightful comment concerning the reflectivity of Mo and the potential for significant power loss in our waveguide design. The waveguide serves as a transmitter that delivers light generated by an external source to the target location. Unlike typical biodegradable self-emitting devices such as micro-LEDs^{R1}, waveguides enable external control of power supply and heat cooling, allowing for the maintenance of high output levels from the external light source to deliver light at the desired intensity. The transmission loss of the optical waveguide can be compensated for by controlling the output of the external light source. Nevertheless, extremely high transmission losses prevent the device from maintaining its intrinsic functionality. To elucidate the efficiency and feasibility of the proposed system, it is necessary to consider the effectiveness and losses of the waveguide. Therefore, we first conducted additional measurements on the intensity of the light emitted from the tip of the waveguide. The intensity of the light transmitted through the PLGA waveguide was quantified to verify its effectiveness in delivering light for stimulating the target tissue. Furthermore, to enhance the reliability and precision of our device, we have additionally manufactured a headstage that was not previously implemented. The created headstage seamlessly connects the optical cannula with the device. When the output power of the optical cannula is 26.4 mW, the average light intensity emitted from the end of the waveguide to stimulate target tissues 63.06 mWmm^{-2} . This exceeds the previously mentioned reference value, thereby being sufficient to induce adequate stimulation.

Next, to assess the amount of light absorbed by Mo reflective layer throughout the guiding, we first considered a single-channel waveguide system shown in **Fig. S7a**. The proposed waveguide utilizes not just single incident angle but all possible angles for waveguiding, thus we discretized the angles capable of total internal reflection (TIR) and conducted ray tracing simulations. We simulated a single channel PLGA waveguide with light sources satisfying TIR condition, by controlling incident angle from 0° to 19° . In this system, the loss of waveguide is only attributed to Mo absorption because neither light is absorbed by the bottom Si layer nor leaks out of the waveguide. The waveguide losses for discretized angles, including the cumulative sum of each case, are presented in **Fig. S7b**. With larger incident angle, the number of reflections on the Mo layer rises, leading to increased absorption loss. Cumulative loss in the waveguide length of 2cm is -6.72dB meaning a loss per unit length of 3.36dB/cm. Afterward, we performed simulations with the same waveguide utilizing non-discretized fiber source to assess how much the reflection on Mo layer occurs during propagation. **Fig. S7c** shows the result of the number of rays and cumulative power relative to the number of reflections on the Mo layer. The inset illustrates how the number of reflections varies for rays starting at different incident angles. Substantial amount of light reached the destination after undergoing a few reflections on the Mo layer while retaining its power. Considering all light propagating through the single-channel waveguide, the final delivered power is 24.98% of input.

Supplementary Figure S7 | The single-channel waveguide simulation with Mo reflective layer. (a) Simulation system to measure loss of single-channel waveguide. The waveguide consists of PLGA waveguide, PLGA substrate, Mo reflective layer and Si nanomembrane. The inset below depicts rays which propagate within the waveguide at various angle. (b) The loss of each discretized angle and cumulative loss. A larger incident angle results in great increased losses due to absorption. The cumulative loss is calculated by adding up the power contributions of each angle. (c) The number of rays in relation to the number of reflections on Mo. The inset illustrates how the number of reflections varies for rays starting at different incident angels.

Subsequently, we conducted simulations for the proposed system shown in **Fig. S8a**. Seven power detector, positioned at 1.5mm intervals, measure the transmitted power at each location to calculate propagation loss. Since only two longer channels reach the 7th detector, we extended two shorter channels through dummy waveguide. **Fig. S8b** shows result of guided power measured from detectors. Initially, 20% of light is leaked through the PLGA substrate and the light either exits the system resulting loss or be guided through the substrate. The absorption loss caused by Mo reflective layer does not occur from the starting point of the waveguide but occurs from the point where the waveguide and Mo reflective layer overlap. As the waveguide overlapping with the Mo reflective layer, linear absorption loss is measured at -7.059dB/cm. Note that the loss is less than real value due to additional loss coming from the length compensation. As mentioned earlier in single-channel waveguide where power remains quite substantial, although the absorption by the Mo reflective layer is not negligible, a significant amount of power is still transmitted due to diverse angles where light can propagate within the waveguide. The loss is sufficiently acceptable because our system is designed for optical propagation at distances less than 1cm.

Supplementary Figure S8 | The actual 4-channel waveguide simulation with Mo reflective layer. (a) Simulation system and detector arrangement for loss measurement. The blue segment at the end of waveguide is an extension of short waveguide to compensate the length mismatch. (b) Loss of the compensated waveguide. Absorption losses occur after where the waveguide overlaps with the Mo layer, indicated green area.

[R1] Lu, D., Liu, T. L., Chang, J. K., Peng, D., Zhang, Y., Shin, J., ... & Rogers, J. A. (2019). Transient Light-Emitting Diodes Constructed from Semiconductors and Transparent Conductors that Biodegrade Under Physiological Conditions. *Advanced Materials*, 31(42), 1902739.

Our modification to the manuscript:

There is a modification in the contents of the sentence.

Line 294: At the bottom side of waveguide, reflection occurs at the interface between the PLGA substrate and electrode array, which minimizes an artifact of Si electrode. Considerable absorption loss is present since reflection from metal inevitably accompanying absorption. Nevertheless, the proposed waveguide delivers a substantial power enough to stimulate neurons (Supplemental Figure S7 and S8). The guided light is transmitted to each electrode and undergoes reflections once again at the inclined waveguide tips.

Line 496: Upon exposure to 460nm pulses transmitted from a waveguide (input power: 24 mW, stimulation intensity: 63.08 mWmm^{-2}), Chr2 ion channels within neurons in a specific target region of the cerebral cortex of the transgenic mouse are depolarized without overheating the adjacent local tissue, leading to the generation of action potentials and subsequent activation of neurons (Supplemental Figure S34).⁶⁴

Line 642:

Ray-tracing simulation

To perform a comprehensive 3D ray-tracing simulation based on the Monte-Carlo method, commercial software called OpticStudio 16.0 by ZEMAX, Inc. was utilized. The simulation employed a monochromatic point light source with a cone angle of 19.5° , inducing TIR in the

waveguide (Supplemental Figure S8), located at the entrance of the waveguide structure. The refractive index of the waveguide was set to 1.47, and for increased accuracy, the refractive indices of the Si and metal were obtained from measured results.^{65,66} The simulation used 1×10^7 rays to obtain stable calculation results, and a rectangular detector with dimensions of 2500×500 pixels in the x and y directions was employed to record the light propagation from the entrance to the tip of the waveguide. 2D light distributions were reconstructed using MATLAB software by Mathworks Inc.

Path analysis in OpticStudio is conducted to specify absorption loss at the anti-artifact layer. The result of the path analysis is graphically represented using Origin 2022 by OriginLab Corporation.

There is a deleted sentence in the contents.

Line 360: ~~Apart from the tip angle, the presence of a reflection layer below the bottom side of the waveguide also significantly affects delivery of light(Supplemental Figure S10). When the waveguide lacks a reflection layer and consists only Si, the majority of the light is absorbed by a Si electrode. On the other hand, a waveguide tip with a metal reflector emits significantly more light than the waveguide tip without a reflector owing to a light loss minimization. The detailed procedure for the ray tracing simulation is described in Methods section.~~

There is an additional supplementary figure.

Supplementary Figure S7 | The single-channel waveguide simulation with Mo reflective layer. (a) Simulation system to measure loss of single-channel waveguide. The waveguide consists of PLGA waveguide, PLGA substrate, Mo reflective layer and Si nanomembrane. The inset below depicts rays which propagate within the waveguide at various angle. (b) The loss of each discretized angle and cumulative loss. A larger incident angle results in great increased losses due to absorption. The cumulative loss is calculated by adding up the power contributions of each angle. (c) The number of rays in relation to the number of reflections on Mo. The inset illustrates how the number of reflections varies for rays starting at different incident angles.

Supplementary Figure S8 | The actual 4-channel waveguide simulation with Mo reflective layer. (a) Simulation system and detector arrangement for loss measurement. The blue segment at the end of waveguide is an extension of short waveguide to compensate the length mismatch. (b) Loss of the compensated waveguide. Absorption losses occur after where the waveguide overlaps with the Mo layer, indicated green area.

Major comment #5: In Fig. 5, Thy-1:ChR2 mice are used. Fluorescence images should be provided to illustrate the expression of ChR2 in the targeted brain region.

Our response: We verified the fluorescence images of Thy-1:ChR2 mice by carrying out the histology experiments. The fluorescence images were illustrated in Supplemental Figure S31.

Our modification to the manuscript:

There is an additional supplementary figure.

Supplementary Figure S31 | Fluorescence images of cortical area from Thy-1: ChR2 transgenic mice (YFP). Cell nuclei are visualized with DAPI stain (blue). (scale bars = 500 μ m)

Major comment #6: The paper only focuses on examining the acute performance of the neural implant system, both in vitro and in vivo. As the materials gradually degrade, it can

substantially impact long-term performance. For instance, the swelling of PLGA can significantly influence optical properties. The dissolution of Si and Mo can introduce recording artifacts and impair stimulation efficiency. Also, chronic degradation analysis should be performed *in vivo*, to demonstrate the device's disappearance in the mouse brain.

Our response: We appreciate the reviewer for raising this important comment. We conducted two additional experiments to verify the operational lifespan and residual lifespan of the device in a state of chronic *in vivo* implantation. After fully implanting the device into the cerebral cortex of transgenic mice through craniotomy, regular assessments were conducted to ensure that the device was operating as intended. These evaluations included monitoring the device's functionality, stability, and performance over an extended period. First, we observed the duration for which the device consistently maintained its initial functionality and performance. For chronic implantation in transgenic mice, we customized the headstage to integrate the optical cannula with the device. Subsequently, we conducted optogenetic stimulation and LFP recording in mice with chronic implants. The device maintained its initial performance and operated in a stable manner, recording similar evoked LFPs under the same stimulation conditions for more than 14 days following the initial implantation. Starting with malfunctions of the device on day 20, several channels exhibit signs of malfunction by day 21, due to the occurrence of an open-circuit state of the electrodes caused by bioresorption. Furthermore, the remaining operational channels exhibit reduced light transmission efficiency due to the *in vivo* biodegradation and erosion of the PLGA waveguide, leading to a reduction in the peak amplitude of the recorded LFPs to below 150 μ V attributed to the diminished light intensity for stimulation. Consequently, all channels failed by 28 days post-implantation.

Next, to further evaluate the biodegradable kinetic of the device, we utilized CT scanning. For 7 weeks, we observed the gradual breakdown and absorption of the device material, confirming its biodegradable nature. Through these combined efforts, we were able to determine both the functional lifespan of the device and its biodegradation duration during chronic implantation. The implanted device was traceable on day 21, after which it completely dissolved by the day 50. In summary, these results indicate that the fully implanted bioresorbable hybrid device operates temporarily while maintaining performance, successfully records evoked LFPs, and ultimately dissolves and disappears within the body.

Our modification to the manuscript:

There is a modification in the contents of the sentence.

Line 506: **Fig. 5f** display the evoked LFPs recorded from each electrode channel of the bioresorbable hybrid system, induced by 460 nm pulses (intensity: 63.08 mWmm⁻², duration: 30 ms, frequency: 2 Hz stimulation). On day 0, in contrast to the control group of wild-type mice, the peak of the optogenetically-induced evoked LFPs in the cerebral cortex of Thy-1: ChR2 transgenic mice occurred approximately 15 ms after initiation of stimulation, with magnitudes ranging from 200 to 250 μ V (Supplemental Figure S35 and S36). Additionally, as the stimulation intensity increased from 15.77 mWmm⁻², 31.54 mWmm⁻², to 63.08 mWmm⁻², evoked LFP peaks of 100 μ V, 160 μ V, and 230 μ V, respectively, were observed (Supplemental Figure S37). The device operates in a stable manner without any significant change in performance, recording similar evoked LFPs in the same stimulation condition for 14 days following the initial implantation. Starting with malfunctions of the device on day 20, several

channels exhibit signs of malfunction by day 21, due to the occurrence of an open-circuit state of the electrodes caused by bioresorption.³⁴ And the remaining operational channel exhibit a decrease in light transmission efficiency due to the *in vivo* degradation and erosion of the PLGA waveguide, leading to a reduction in the peak amplitude of the recorded LFPs to below 150 μV attributed to the diminished light intensity for stimulation.⁶⁵ Consequently, all channels had failed by the 28 days post-implantation. Additionally, the dissolution process of the implanted device was monitored using computed x-ray tomography. The implanted device was traceable on day 21, after which it completely dissolved by the day 50 (Supplemental Figure S38). Collectively, these results suggest that the fully implanted bioresorbable hybrid device operated transiently while maintaining its performance, successfully recording evoked LFPs with minimal artifacts and simultaneously transmitting light from a laser source for optogenetic stimulation, ultimately dissolving and disappearing *in vivo*.

There is a modification in the figure 5.

There is an additional supplementary figure.

Supplementary Figure S38 | Dissolution characteristic of the implanted device. Computed tomography images of coronal section of the mouse skull collected over 49 days following the device implantation. The white dotted box highlights the bioresorbable device.

Minor comment #1: Fig. 3b and 3c, what are the differences between n-Si and p-Si in this

case? More discussions should be added.

Our response: We thank the reviewer for this comment. In n-type and p-type materials, the major carriers differ, leading to band bending in opposite biases at the Fermi energy level equilibrium, particularly when the Bequerel effect occurs. At the interface between p-type silicon and an electrolyte, a current is generated in the opposite direction compared to n-type materials. Furthermore, due to the difference in carrier mobility, n-type silicon exhibits lower resistivity at the same doping concentration. This results in a higher Signal-to-Noise Ratio (SNR) for low-frequency signals and reduced signal transmission loss.

Our modification to the manuscript:

There is a modification and addition in the contents of the sentence.

Line 256: The conductive electrode is composed of a Mo and highly P-doped Si nanomembrane bilayer (300/300 nm). At the same doping concentration, n-type Si comprising the electrode has a relatively lower resistivity due to its electrons, the major carriers, having higher mobility compared to the holes in p-type Si. This results in minimized low-frequency signal loss in electrodes and transmission lines.

Line 387: In the same manner, p-type Si is induced a reverse bias opposite to n-type Si (Supplemental Figure S19).

There is an additional supplementary figure.

Supplementary Figure S18 | Energy band diagrams at the p-Si electrode and biofluid interface under (a) equilibrium and (b) illuminated conditions.

Minor comment #2: Fig. 1i, why do the authors choose a solution with pH = 4.01? The brain environment should be weak base solution with pH = 7.4. To accelerate the degradation, using a solution with pH > 7 will be better.

Our response: We thank the reviewer for delicate comments. Since the internal body environment is slightly alkaline, we conducted acceleration tests in a pH 9.87 buffer instead of the acidic solution.

Our modification to the manuscript:

There is a modification in the figure 1.

Minor comment #3: Fig. 2b, how to form the waveguide with facets with different angles? Fabrication details are needed.

Our response: Thanks for this comment. When making master molds, we use UV-curable epoxy (SU-8). Using a jig with a slit through 3D printing, it is manufactured by inserting a substrate into the slit to set an angle and then irradiating it with UV light. The manufacturing process of the master mold is as follows in **Figure R1**.

Figure R1. Fabrication process of PLGA waveguide master mold.

Our modification to the manuscript:

There is a modification and addition in the contents of the sentence.

Line 559:

PLGA waveguide fabrication

Fabrication of the poly(lactic-co-glycolic acid)(PLGA) waveguide start with fabrication of master mold. On the prepared glass substrate, polyimide (PI) (~4 μm) is spun coated, baked (150 °C for 10 min, 210 °C for 180 min) and Cu (~300 nm) is deposited by sputtering. After patterning the waveguide pattern through photolithography, PI and Cu are etched, sequentially. And spin-casting SU-8 100 (~100 μm) on a glass substrate. Photoexposure of the SU-8 100 layer after soft baking (65 °C for 20 min, 95 °C for 90 min) was carried out for patterning waveguides in a state where the glass substrate was turned over and fixed jig to tilt the master

mold at 60°(Supplemental Figure SX). After post exposure baking (65 °C for 1 min, 95 °C for 15 min) and SU-8 100 development, the fabricated master mold was fixed on a petri dish and 5:1 polydimethylsiloxane (PDMS, Dow Corning) was poured to obtain a PDMS mold. The completely cured PDMS mold was separated from the master mold and petri dish before being fixed on another glass substrate with quicksilver to prevent deformation of the mold. A fabricated PLGA-chloroform solution (5%w/v) (PLA:PGA=75:25, Rimless Industry Co., Mw=10,000 g/Mol, IV=0.85 dL/g) was poured over the PDMS mold and dried at room temperature to form the PLGA waveguide film. The fabricated dry film was not separated from the PDMS mold before electrode array transfer printing.

There is an additional supplementary figure.

Supplementary Figure S13 | Photographs of the jig for making the master mold. (a) Master mold with 60° inclined slits. (b) Photograph of UV irradiation on the master mold fixed to the jig.

Minor comment #4: Fig. 3e uses the unit of mW, while Line 367 uses the unit of mW/cm². These units should be consistent.

Our response: We thanks the reviewer for sharp comment. To avoid confusing the reader, we have consistently noted the units.

Our modification to the manuscript:

There is a modification and addition in the contents of the sentence.

Line 410: To evaluate light-induced artifacts, 460 nm laser pulses (core diameter: 105um, duration: 100ms, frequency: 2Hz) were applied to each electrode while gradually increasing the light intensity from 126.1 mWmm⁻² to 2522.5 mWmm⁻² through an optical fiber fixed perpendicular to the stereotaxis (Supplemental Figure S19).

Line 420: The artifacts with peaks from 400μV to 3mV were induced as the light intensity increased from 126.1 mWmm⁻² to 2522.5 mWmm⁻² and at higher light intensities, the artifacts reached a saturation state (Supplemental Figure S23).

Line 423: In contrast, when the Mo/Si bilayer electrode was illuminated at intensities below 630.6 mWmm⁻², low levels of noise without any artifacts were observed. Negligible artifacts were observed below peak level of 200 μV with electrodes exposed to light intensities below

2522.5 mWmm⁻².

Line 430: Blue light of intensity 126.12 mWmm⁻² was irradiated on the stimulation electrode, and blue light of intensity 73.15 mWmm⁻² passed through the electrode to stimulate the tissue.

There is a modification in the figure 3.

Minor comment #5: Fig. 3f, the control electrodes also exhibit different levels of artefacts. Why?

Our response: We thank the reviewer for your insightful comment. Artifacts become saturated as the intensity of light increases. In the case of manufactured Si electrodes, saturation occurs when light with an intensity of approximately 2500 mWmm⁻² or more is applied. We used a higher intensity laser, providing additional data beyond the saturation point.

Our modification to the manuscript:

There is a modification and addition in the contents of the sentence.

Line 410: The artifacts with peaks from 400μV to 3mV were induced as the light intensity increased from 126.1 mWmm⁻² to 2522.5 mWmm⁻² and at higher light intensities, the artifacts reached a saturation state (Supplemental Figure S23).

There is an additional supplementary figure.

Supplementary Figure S23 | Photoinduced artifact on monolayer Si nanomembrane electrodes. The Si electrode was subjected to laser pulses with intensities of (a) 2522.48 mWmm⁻² and 3782.72 mWmm⁻².

REVIEWER COMMENTS

Reviewer #1 (Remarks to the Author):

All of my comments have been diligently addressed by the authors, and I am confident that the revised manuscript is now suitable for publication in its current form.

Reviewer #2 (Remarks to the Author):

General comment:

The manuscript by Cho et al. has shown considerable improvement upon revision. The authors have provided a comprehensive set of material characterizations, effectively supporting their approach towards the stability and optical properties of the devices discussed. However, there are a few minor points that need to be addressed before considering acceptance.

Comment 1:

It would be beneficial for the manuscript if the authors could provide the standard deviation for the refractive index measurements of PLGA polymer films, particularly if $n=3$. Also, please provide standard deviation for Supplementary Fig S2.

Comment 2:

Please specify what methods were used to determine the GFAP and Iba1 fluorescence intensity. Specifically, it is important to clarify whether the exposure parameters were consistent across all individual images. Including references for these methods might also aid in providing the audience with a better understanding.

Comment 3:

Please specify the methods used for this sorting in the Methods section, including appropriate citations. This will provide clarity on the techniques used and support the reproducibility of the research.

Reviewer #3 (Remarks to the Author):

In the revised manuscript, the authors provide additional results in response to the Reviewers' comments. However, concerns still remain within these results.

- (1) The authors provide new results (Fig. S7 and S8) to address the issues related to the waveguide loss. According to Fig. S7, the calculated results is about 3 dB/cm, and specifically about 6 dB/cm for light with an angle of 19° .

Let us do a simple calculation to disprove this result:

The waveguide thickness is about 100 μm . For light at 19° , the propagation length is $100 \mu\text{m} / \tan(19^\circ) * 2 = 600 \mu\text{m} = 0.6 \text{ mm}$, for one single reflection at the Mo surface. The reflection loss at the Mo surface is about 50% (3 dB) at the wavelength of 460 nm. The 3 dB point is at 0.6 mm, not 0.6 cm shown in Fig. S7b. Therefore, the actual waveguide loss is about 60 dB/cm, not 6 dB/cm. The loss is significantly underestimated. Actually, for a waveguide with a length of 2 cm, only 10^{-12} of the incident power can reach the waveguide tip.

In light of these major miscalculation, the entire waveguide design is fundamentally flawed.

(adapted from Fig. S7c)

- (2) In Fig. 5f, the four channels record almost exactly the same electrical signals under optical stimulation. It is obvious that the four electrodes are placed on different brain regions and should have recorded somewhat different signals. Therefore, these results are doubtful and highly likely to be artefacts.

f.

Day 0

* : Optogenetic stimulation

In addition, the authors did not perform the experiments I suggest in the previous comment *“only illuminate one or two waveguide channels, and measure the LFP signals from all the four electrode channels”*.

To summarize, I cannot recommend the publication of this work, unless the authors can prove that the above waveguide calculation is wrong, and clear the doubts of the recorded electrical signals.

Reviewer #3

General Comment: In the revised manuscript, the authors provide additional results in response to the Reviewers' comments. However, concerns still remain within these results.

Our response: We thank the reviewer for this comment. We made our revision with pleasure based on the reviewer's opinion, and all the details of our modifications are indicated in Our response.

Comment #1: The authors provide new results (Fig. S7 and S8) to address the issues related to the waveguide loss. According to Fig. S7, the calculated results is about 3 dB/cm, and specifically about 6 dB/cm for light with an angle of 19°.

Let us do a simple calculation to disprove this result:

*The waveguide thickness is about 100 μm . For light at 19°, the propagation length is $100 \mu\text{m} / \tan(19^\circ) * 2 = 600 \mu\text{m} = 0.6 \text{ mm}$, for one single reflection at the Mo surface. The reflection loss at the Mo surface is about 50% (3 dB) at the wavelength of 460 nm. The 3 dB point is at 0.6 mm, not 0.6 cm shown in Fig. S7b. Therefore, the actual waveguide loss is about 60 dB/cm, not 6 dB/cm. The loss is significantly underestimated. Actually, for a waveguide with a length of 2 cm, only 10-12 of the incident power can reach the waveguide tip.*

In light of these major miscalculation, the entire waveguide design is fundamentally flawed.

Our response: We are grateful the reviewer's assistance in identifying a miscalculation in our work. Our methodology involved employing a set of detectors at specific intervals to measure the delivered power within the proposed waveguide. To derive linear loss from discretized losses, we used line fitting. Unfortunately, an error occurred during this process: an incorrect value was inputted for the 'waveguide length' axis in our fitting procedure. The corrected loss line and discretized loss at a 19° incident angle are illustrated in **Fig. R1a**. We sincerely apologize for this error, which could have led to misleading conclusions, and have replaced the erroneous **Fig. S7b** with **Fig. R1b**.

Figure R1. The corrected loss of the proposed waveguide in condition at the incident angle of 19° and all incident angles. (a) The representative loss at the incident angle of 19°. Blue spheres

mean the losses for each detector. Vertical gray lines in the background correspond to the point where rays are reflected from the Mo reflective layer. The number of reflections between each discrete detector affect to the loss. Pre-corrected and post-corrected losses are indicated with red colored solid and dashed lines, respectively. (b) Correctly fitted loss line in accordance with variation of the incident angle. Losses increased in all incident angles. The cumulative loss remains consistent with that of pre-corrected version, because it is derived not through uncorrected line fitting but rather by summing discretized losses at each angle.

Despite correcting the underestimated loss, a discrepancy remains between the simple calculation loss provided by the reviewer and our single-channel simulation results. We attribute this difference to the incident angle on the Mo layer. **Fig. R2a** demonstrates how absorptance varies with the incident angle to Mo layer. At normal incidence to Mo, the absorptance accounts for 52% of the incident power (**Fig. R2a**, top). However, for oblique incidence in the proposed waveguide, the absorptance of Mo is lower due to the oblique incident angle θ_{Mo} compared to the normal incident case (**Fig. R2a**, bottom). The θ_{Mo} is determined by the θ_{WG} , which is an initial angle at entrance of the waveguide.

Figure R2. The changing absorption of Mo with the incident angle. (a) Schematic of normal incidence (top) and oblique incidence in the proposed waveguide (bottom). The absorption rate changes with the θ_{Mo} , that is determined by the θ_{WG} at entrance of the waveguide. (b) Absorption spectra for θ_{WG} . In all possible range of incident angle (top), normal incidence to the Mo layer is indicated by the blue dot. The range used in the simulation, which is left side by the violet dashes, is plotted separately (bottom). (c) Losses of waveguide for two polarization states and unpolarized state. Simulation was conducted in a single-channel waveguide with θ_{WG} of 19° . The inset image shows absorption loss for a single reflection in both polarizations, respectively. (d) Power distribution of optical fibers. The low-order modes at the center occupy substantial amount of power and deliver considerable power.

Fig. R2b illustrates the absorption spectra corresponding to various θ_{WG} in a single reflection. Absorptance for unpolarized light is calculated as the average of absorptance for TE and TM polarization. At θ_{WG} of 90° , indicating normal incidence to Mo, the absorptance is approximately 52%, which is generally known as the absorptance for Mo regardless of polarization (**Fig. R2b**, top). However, within θ_{WG} range used to induce total internal reflection (TIR) at the interface of the PLGA waveguide and cerebrospinal fluid, the absorptance varies significantly depending on polarization (**Fig. R2b**, bottom). Notably, at a θ_{WG} of 19° , the absorptance differs significantly between two polarizations: approximately 82% for TM polarized light and only 18% for TE polarized light. The observed higher absorption in TM-polarized light appears to be due to the ohmic losses of a TM guided mode. This is a result of the resonant coupling of the guided mode to the lossy surface plasmon polariton, which is supported by the thin metal film [R1]. Conversely, as the TE mode is not affected by this coupling [R2], absorption by the Mo layer is minimized. The larger TM to TE loss ratio in a planar optical waveguide with metal has been utilized as a TE mode pass polarizer [R1, R2].

Waveguide loss for both polarization states and an unpolarized state is depicted in **Fig. R2c**. The inset displays the absorbed powers, in decibels, for a single reflection at θ_{WG} of 19° . The losses for a single polarization align with the absorption rate presented earlier in **Fig. R2b**. TM polarization shows a huge loss (*i.e.*, 7.41dB/reflection corresponding to absorption of 82%) at intervals of reflection at Mo layer, while TE polarization incurs a loss of 1.06dB/reflection (18% absorption). Meanwhile, the loss for an unpolarized light, derived from the average absorptance of TE and TM polarization, initially increases sharply due to the rapid loss in TM polarization. Subsequently, as the loss of TM polarization becomes negligible, the overall loss follows the gradual decrease observed in TE polarization. As a result, the loss at a θ_{WG} of 19° is -20dB at 1cm and -35dB at 2cm. As depicted in **Fig. R2b**, a lower θ_{WG} corresponds to lower losses. Considering the power distribution in optical fibers, as illustrated in **Fig. R2d**, low-order modes near the fiber center carry more power than high-order modes [R3]. Since the light coupled into our proposed waveguide predominantly consists of these low-order (*i.e.*, low θ_{WG}) modes, our proposed waveguide is capable of delivering a prominent amount of light.

Reference:

- [R1] V. K. Sharma, Anil Kumar, A. Kapoor, “High extinction ratio metal-insulator-semiconductor waveguide surface plasmon polariton polarizer”, *Optics Communications* 284, 7 (2011)
- [R2] Ruo-Zhou Li, Li-Jiang Zhang, Wei Hu, Long-De Wang, Jie Tang, and Tong Zhang, “Flexible TE-pass polymer waveguide polarizer with low bending loss”, *IEEE Photonics Technology Letters* 28, 22 (2016).
- [R3] Senior, John M., and M. Yousif Jamro. *Optical fiber communications: principles and practice*. Pearson Education, 2009.

Our modification to the manuscript:

There is a modification and addition in the contents of the sentence.

Line 297: Nevertheless, the proposed waveguide delivers a substantial power enough to stimulate neurons (Supplemental Figure S7, S8 and S9).

There is a modification in the supplementary figures.

Supplementary Figure S7 | The single-channel waveguide simulation with Mo reflective layer.

(a) Simulation system to measure loss of single-channel waveguide. The waveguide consists of PLGA waveguide, PLGA substrate, Mo reflective layer and Si nanomembrane. The inset below depicts rays which propagate within the waveguide at various angle. (b) The loss of each discretized angle and cumulative loss. A larger incident angle results in great increased losses due to absorption. The cumulative loss is calculated by adding up the power contributions of each angle. (c) The number of rays in relation to the number of reflections on Mo. The inset illustrates how the number of reflections varies for rays starting at different incident angles.

Supplementary Figure S8 | The changing absorption of Mo with the incident angle. (a)

Schematic of normal incidence (top) and oblique incidence in the proposed waveguide (bottom). The absorption rate changes with the θ_{Mo} , that is determined by the θ_{WG} at entrance of the waveguide. (b) Absorption spectra for θ_{WG} . In all possible range of incident angle (top), normal incidence to the Mo layer is indicated by the blue dot. The range used in the simulation, which is left side by the violet dashes, is plotted separately (bottom). (c) Losses of waveguide for two polarization states and unpolarized state. Simulation was conducted in a single-channel waveguide with θ_{WG} of 19° . The inset image shows absorption loss for a single reflection in both polarizations, respectively. (d) Power distribution of optical fibers. The low-order modes at the center occupy substantial amount of power and deliver considerable power.

Comment #2: In Fig. 5f, the four channels record almost exactly the same electrical signals under optical stimulation. It is obvious that the four electrodes are placed on different brain regions and should have recorded somewhat different signals. Therefore, these results are doubtful and highly likely to be artefacts. In addition, the authors did not perform the experiments I suggest in the previous comment “only illuminate one or two waveguide channels, and measure the LFP signals from all the four electrode channels”.

Our response: We thank the reviewer for this comment. First, we conducted an in vivo experiment with a control group to verify whether the recorded signals were artifacts induced by pulse light or local field potentials (LFPs) originating from neural activities. After implanting our device into the cortical region of Thy-1:ChR2 transgenic mice and wild-type mice as a control, we subsequently recorded electrical signals induced through optical stimulation using 460 nm pulses (intensity: 63.08 mWmm^{-2} , duration: 30 ms, frequency: 2 Hz stimulation). As shown in **Fig. R3**, the signals recorded by the black line represent electrical signals recorded in transgenic mouse, while the signals recorded by the red line represent electrical signals recorded in wild-type mouse. Both experiments were conducted under controlled environmental conditions, utilizing identical device settings.

Figure R3. ECoG including evoked LFPs recorded by 4-channel electrode array for 4-ch pulsed photostimulation. Recorded LFP signal from **a**, Thy-1: ChR2 mouse and **b**, wild-type mouse. The blue line indicates the instant of stimulation.

Nevertheless, there is a distinct difference in the signals recorded from each device. If the signals recorded in **Fig. 5f** of the manuscript were photoinduced artifacts, similar forms of electrical signals should have been recorded in the wild-type mouse as a control conducted under the same conditions. However, when the cortex was stimulated with 460nm pulses, a

sinusoidal-shaped electrical signal was exclusively recorded in transgenic mice, indicating a LFPs arising from neural activities induced by optical stimulation.

Additionally, in response to the reviewer's suggestion, after implanting the device into the cortical region of transgenic mice, optical stimulation (intensity: 63.08 mWmm^{-2} , duration: 30 ms, frequency: 1 Hz stimulation) was applied exclusively to a single electrode (2nd channel) on the electrode array. The results of the corresponding in vivo experiment are presented in **Fig. R4**. As shown in **Fig. R4a**, when optical stimulation is applied to the second electrode spot, optogenetically induced LFPs are directly recorded in the channel-2. Furthermore, the induced LFPs propagate, attenuating as they spread to neighboring electrodes and are sequentially recorded. If the recorded electrical signals were artifacts induced by photo stimulation, there would have been no signals recorded on the remaining electrodes. Even if light scattering caused artifacts on neighboring electrodes, we would have observed immediate signals on the remaining electrodes. However, as shown in **Fig. R4b**, the induced LFPs show an approximate 2 ms delay before being recorded at the farthest channel-4, located 1 mm away.

Figure R4. ECoG including evoked LFPs recorded by 4-channel electrode array for 1-ch pulsed photo-stimulation via optical fiber. **a**, Recorded LFP signal from Thy-1: Chr2 mouse. Stimulation is applied exclusively at the electrode of channel 2. The blue line indicates the instant of stimulation. **b**, Sorted optogenetic LFP responses recorded from the 4-ch electrode array. The stimulation occurs at 0 ms.

In summary, we simultaneously stimulated four areas to demonstrate the potential for multi-channel and large-area stimulation. The recorded neural signals, especially LFPs, influenced surrounding electrodes, leading to similar LFPs being captured across all four electrodes due to mutual interference. Nevertheless, as depicted in Figs. R3 and R4, it is clear that the recorded signals are not artifacts induced by light, but rather LFPs resulting from optical stimulation. Furthermore, soft lithography allows for the flexible adjustment of waveguide shapes, enabling the customization of channels with specified positions and numbers for selective illumination of target areas. So, our future work will focus on achieving highly selective stimulation integrated with large-area, high-density electrode arrays in pathological models.

Our modification to the manuscript: none

Thank you very much again for your insightful comments. We feel that these comments have helped to improve the quality of the manuscript significantly.

REVIEWERS' COMMENTS

Reviewer #3 (Remarks to the Author):

I appreciate that the authors provide clear explanations on the waveguide loss and modify their calculations. The point on the TE and TM modes at oblique angles is correct.

In addition, they provide additional experiments to validate the optogenetic effects (Figure R4 in their response). These results are excellent and should appear in the paper (main text or supplement).

Other than that, I do not have any further comments. The modified paper can be accepted.

Reviewer #3

General Comment:

I appreciate that the authors provide clear explanations on the waveguide loss and modify their calculations. The point on the TE and TM modes at oblique angles is correct.

In addition, they provide additional experiments to validate the optogenetic effects (Figure R4 in their response). These results are excellent and should appear in the paper (main text or supplement).

Other than that, I do not have any further comments. The modified paper can be accepted.

Our response:

First, we thank the reviewer for this positive comment, and the recommendation to publish in *Nature Communications*. We agree with the reviewer's request to add results of additional experiments to validate the optogenetic effects. So we have added discussion and supplemental figure in our manuscripts about single channel stimulation experiment.

Our modification to the manuscript:

There is a modification in the contents of the sentence.

Line 434: Furthermore, in Thy-1:ChR2 transgenic mice, optical stimulation (intensity: 63.08 mWmm⁻², duration: 30 ms, frequency: 1 Hz) was exclusively applied to a single electrode, and LFPs induced by photo-stimulation were recorded directly on the stimulated electrode (Supplemental Figure S27).

There is an additional supplementary figure.

Supplementary Figure S27 | ECoG including evoked LFPs recorded by 4-channel electrode array from single channel pulsed photo-stimulation via optical fiber on transgenic mice. a, Recorded LFP signal from Thy-1: ChR2 mouse. Stimulation with 460 nm pulses (intensity: 63.08 mWmm⁻², duration: 30 ms, frequency: 1 Hz) is applied exclusively at the electrode of channel 2. The blue line indicates the instant of stimulation. **b,** Sorted optogenetic LFP responses recorded from the 4-ch electrode array. The stimulation occurs at 0 ms.